# The spinal cord facilitates cerebellar upper limb motor learning and control; inputs from neuromusculoskeletal simulation

**Alice Bruel** [1]☉*, **Ignacio Abadía** [2]☉*, **Thibault Collin** [3], **Icare Sakr** [3], **Henri Lorach** [3], **Niceto R. Luque** [2], **Eduardo Ros** [2], **Auke Ijspeert** [1]

**1** Biorobotics Laboratory, EPFL, Lausanne, Switzerland, **2** Research Centre for Information and Communication Technologies, Department of Computer Engineering, Automation and Robotics, University of Granada, Granada, Spain, **3** NeuroRestore, EPFL, Lausanne, Switzerland

☉ These authors contributed equally to this work.
* alice.bruel@epfl.ch (AB); iabadia@ugr.es (IA)

## Abstract

Complex interactions between brain regions and the spinal cord (SC) govern body motion, which is ultimately driven by muscle activation. Motor planning or learning are mainly conducted at higher brain regions, whilst the SC acts as a brain-muscle gateway and as a motor control centre providing fast reflexes and muscle activity regulation. Thus, higher brain areas need to cope with the SC as an inherent and evolutionary older part of the body dynamics. Here, we address the question of how SC dynamics affects motor learning within the cerebellum; in particular, does the SC facilitate cerebellar motor learning or constitute a biological constraint? We provide an exploratory framework by integrating biologically plausible cerebellar and SC computational models in a musculoskeletal upper limb control loop. The cerebellar model, equipped with the main form of cerebellar plasticity, provides motor adaptation; whilst the SC model implements stretch reflex and reciprocal inhibition between antagonist muscles. The resulting spino-cerebellar model is tested performing a set of upper limb motor tasks, including external perturbation studies. A cerebellar model, lacking the implemented SC model and directly controlling the simulated muscles, was also tested in the same. The performances of the spino-cerebellar and cerebellar models were then compared, thus allowing directly addressing the SC influence on cerebellar motor adaptation and learning, and on handling external motor perturbations. Performance was assessed in both joint and muscle space, and compared with kinematic and EMG recordings from healthy participants. The differences in cerebellar synaptic adaptation between both models were also studied. We conclude that the SC facilitates cerebellar motor learning; when the SC circuits are in the loop, faster convergence in motor learning is achieved with simpler cerebellar synaptic weight distributions. The SC is also found to improve robustness against external perturbations, by better reproducing and modulating muscle cocontraction patterns.

**Data Availability Statement:** For reproducibility and comparative purposes, the source code is available on Zenodo at https://doi.org/10.5281/zenodo.7701436.

**Funding:** This work was supported by the following grants and projects: European Union Human Brain Project Specific Grant Agreement 3 (H2020-RIA. 945539), awarded to AI and ER; SPIKEAGE [PID2020-113422GA-I00] by the Spanish Ministry of Science and Innovation MCIN/AEI/10.13039/501100011033, awarded to NRL; DLROB [TED2021-131294B-I00] funded by MCIN/AEI/10.13039/501100011033 and by the European Union NextGenerationEU/PRTR, awarded to NRL; MUSCLEBOT [CNS2022-135243] funded by MCIN/AEI/10.13039/501100011033 and by the European Union NextGenerationEU/PRTR, awarded to NRL. The funders had no role in study design, data collection and analysis, decision to publish, or preparation of the manuscript.

**Competing interests:** The authors have declared that no competing interests exist.

## Summary

Accurate motor control emerges from complex interactions between different brain areas, the spinal cord (SC), and the musculoskeletal system. These different actors contribute with distributed, integrative and complementary roles yet to be fully elucidated. To further study and hypothesise about such interactions, neuromechanical modelling and computational simulation constitute powerful tools. Here, we focus on the SC influence on motor learning in the cerebellum, an issue that has drawn little attention so far; does the SC facilitate or hinder cerebellar motor learning? To address this question, we integrate biologically plausible computational models of the cerebellum and SC, equipped with motor learning capability and fast reflex responses respectively. The resulting spino-cerebellar model is used to control a simulated musculoskeletal upper limb performing a set of motor tasks involving two degrees of freedom. Moreover, we use kinematic and EMG recordings from healthy participants to validate the model performance. The SC fast control primitives operating in muscle space are shown to facilitate cerebellar motor learning, both in terms of kinematics and synaptic adaptation. This, to the best of our knowledge, is the first time to be shown. The SC also modulates muscle cocontraction, improving the robustness against external motor perturbations.

## 1 Introduction

Accurate motor control enables interactions with the environment and others, a process in which sensory information is integrated by the central nervous system (CNS) and translated into muscle activity, eventually driving body motion. Body motion results from the interaction between the musculoskeletal system and diverse neural regions with distributed, integrative and complementary roles [1]. In the brain, various neural regions project descending motor control signals to the spinal cord (SC); e.g., the motor cortex, involved in the volitional control of motion [2]; the basal ganglia, involved in selecting motor behaviour and balance control [3, 4]; the cerebellum, involved in motor coordination and learning [5]. The SC circuits integrate those motor descending signals to regulate motoneuron activity, ultimately driving muscle activation. Besides, the SC also implements its own motor control mechanisms; e.g., fast reflexes, control of rhythmic locomotion movements, or responses against perturbations [5].

Motor control within the CNS could be synthesised as a hierarchical process; higher brain areas govern motor functions such as planning or learning, and the SC then integrates their descending control signals, provides faster and lower-level control mechanisms, and ultimately drives muscle activity. To comprehend and hypothesise about this hierarchical interaction, neuromechanical modelling and computational simulation represent powerful tools, providing a holistic view conjugating from neuron to neural network to motor behaviour levels [6]. To that aim, we present a hierarchical structure comprising: a cerebellar model, a higher brain area equipped with motor learning and adaptation; an SC model, integrating the cerebellar descending control signals and implementing fast-reflexes and muscle activity regulation, and finally actuating a musculoskeletal upper limb model. This spino-cerebellar integration thus provides a computational exploratory framework, which was further complemented with kinematic and EMG data validation. Both the cerebellum and SC main physiological mechanisms have been previously described, however, little attention has been put on the SC influence on cerebellar motor control. Spinal circuits are evolutionary old, they were present in the first vertebrates emerged about 500 million years ago [7] and fully allowed basic locomotion [8]. As new higher neural areas evolved to handle more complex motor control, they had to coexist

and interact with the old lower spinal circuits. It is not clear whether that interaction facilitates motor control or implies a constraint with which higher neural regions have to live with. On the one hand, the SC benefits motor control providing fast feedback loops, lower dimensionality for planning and control, and motor primitives (i.e., low level motor building blocks). On the other hand, higher brain areas have to deal with the highly non-uniform control space and hidden states in the SC, and the need for inverse models that cover not only the body dynamics but also the SC dynamics. Here, we study whether the SC facilitates cerebellar motor learning, or it is simply an evolutionary constraint to be handled.

The cerebellum is key in motor control and coordination, and most importantly motor learning [9]. The Marr-Albus-Ito theory on cerebellar function [10] established the computational principles for supervised cerebellar learning [11], by which the cerebellum enables the adaptation of our actions so their consequences match up to our expectations, i.e., minimising the difference between our intention and the actual movement [12]. This motor learning capability stands upon the plasticity exhibited at the synapses from parallel fibres (PF), i.e., axons of granule cells (GC), to Purkinje cells (PC); plasticity regulated by the action of climbing fibres (CF) reaching PCs [13]. The Marr-Albus-Ito theory assumes the GCs carry a recoding of the sensory inputs conveyed through mossy fibres (MF) [14], whereas CFs carry an instructive signal coding the disparity between our motor expectation and the actual motor state. Despite the well-accepted common ground on the cerebellum established by the Marr-Albus-Ito theory, new findings keep refining the understanding about cerebellar structure and operation, for which computational models are key contributors [15]. Computational models of the cerebellum have been used to study its inner dynamics [16, 17], proving the cerebellar motor learning ability and its capacity to adapt to dynamic changes [18–21]. However, the extensive efforts devoted to cerebellar computational research usually model the cerebellum in isolation. In this work, we build upon and expand previous cerebellar research to include the SC, as theories on the CNS motor function cannot ignore spinal circuitry [22].

Lower down in the CNS hierarchy, the SC transmits control signals from brain motor areas to the muscles, and it also conveys sensory signals from muscle receptors back to the brain. But its role in motor control goes beyond a mere gateway between the brain and muscles [23–25]. The SC contains neural pathways that regulate muscle activity, control reflex responses and produce rhythmic locomotion movements. These spinal pathways channel the sensory feedback mainly from stretch sensitive muscle spindles and tension sensitive Golgi tendon organs (GTO). This sensory feedback is then transmitted to motoneurons through afferent fibres and spinal interneurons, allowing reflex responses and muscle regulation mechanisms: e.g., stretch velocity reflex, static stretch reflex, Golgi tendon reflex, or reciprocal inhibition between antagonist muscles [5]. Besides, these spinal pathways are modulated by higher brain areas during movement execution such as between the stance and swing phases during gait [26, 27], or during arm movements [25, 28, 29], thus highlighting the importance of the interaction between the SC and higher brain areas.

Computational models have been used to gain deeper insight on the SC role in motor control; e.g., control of centre-out reaching movements [22]; control of biceps stretch reflex [30]; reflex modulation via feedback gains [31]; rejection of dynamic perturbations, highlighting the latency hierarchy levels of feedback [32], or the contribution of GTO feedbacks [33]. However, these approaches lacked complex descending signals from higher brain areas, usually applying open-loop supraspinal modules, hence hindering their use to study the interaction between the SC and higher neural regions; larger scale models are required.

Little work has been done on large scale modelling to dig into the SC interaction with higher CNS regions. A recent example coupled spinal circuits with sensory and motor cortex models, forming a feedback control loop designed to reduce the difference between the desired and

perceived state of a planar six-muscle arm [34]. The model showed motor control success and reproduced some previous experimental phenomena, whilst it was suggested that the ataxic nature of the produced movements could be due to the lack of a cerebellum model in the loop.

Regarding spino-cerebellar integration in particular, a few previous computational approaches exist. Contreras-Vidal et al. modelled a cerebellum cooperating with an SC-based muscular force model, together with a central pattern generator representing the motor cortex and basal ganglia [35]. The cerebellar model, developed in analogue form and lacking the temporal correlation nature of cerebellar learning, succeeded in learning muscle synergies, including cocontraction of antagonist pairs, that improved upon the SC feedback control of tracking. Different cerebellar lesions were studied, but the influence of the SC in cerebellar motor adaptation was sidestepped. Subsequently, Spoelstra et al. integrated a cerebellar model with an SC model for postural control of a six-muscle two-dimensional arm model [36]. The study assessed the predictive role of the cerebellum in accurate motor control, but again, the effect of the SC in cerebellar learning was not addressed. More recently, Jo integrated a functional cerebellar model with spinal circuits equipped with plasticity but lacking reflex or other complex spinal dynamics [37]. Results showed the effectiveness of the model to learn movements, with synaptic plasticity at the SC helping to acquire muscle synergies. However, as stated by the author, that learning capacity provided to the SC could be located anywhere in the corticospinal pathway, hence loosening possible conclusions on the cerebellum-SC relation.

With the present work, we intend to extend the spino-cerebellar integration studies; we addressed the questions of whether the SC facilitates cerebellar learning or it is just as an evolutionary constraint, and how the SC contributes to handling motor perturbations. We modelled a biologically plausible cerebellar spiking neural network (SNN), equipped with synaptic plasticity at GC-PC connections guided by the instructive signal conveyed through CFs, thus, able to provide motor adaptation. We added an SC model equipped with stretch reflex and reciprocal inhibition, integrating the descending signals from the cerebellum and sending muscle excitation commands to the musculoskeletal upper limb model, equipped with two degrees of freedom (DOF) actuated by eight Hill-based muscles. Both the cerebellar and SC model were integrated in a negative feedback control loop. The study, developed using computational tools and neuromechanical modelling, is also supported by lab recorded kinematics and EMG data from healthy participants.

In the presented framework, the cerebellar model provides the motor adaptation required for the musculoskeletal upper limb model to achieve a set of goal motor behaviours, i.e., different upper limb movements are defined in joint space (position and velocity), and the cerebellum acquires the inverse model allowing accurate position and velocity tracking. We suggest the SC fast control primitives and regulation of muscle activity to be key in facilitating the cerebellar learning of the muscle dynamics; the SC allowed faster motor learning with simpler cerebellar synaptic adaptation. We also hypothesise that the SC plays a major motor control role through cocontraction modulation; i.e., regulation of simultaneous activation of antagonist muscles. Cocontraction has been shown to improve stability by increasing joint apparent stiffness [38], enhance upper limb movement accuracy [39], and it has also appeared useful in movements requiring robustness against perturbations [40]. We found that the stretch reflex and reciprocal inhibition mechanisms participate in modulating cocontraction, with a significant impact on cerebellar motor adaptation and response against external perturbations.

## 2 Results

We integrated the spinal cord and cerebellum models in an upper limb musculoskeletal feedback control loop (Fig 1A). The spino-cerebellar model commanded the upper limb to

perform a set of motor tasks, a motor benchmark divided in two groups: i) lab recorded upper limb movements performed by two healthy participants to study natural self-selected movements, ii) lab designed upper limb movements with bell-shaped velocity profiles to study standard characteristic reaching movements. A cerebellar model lacking the SC integration performed in the same motor benchmark (Fig 1B) thus providing a spino-cerebellar vs. cerebellar control framework that allowed contextualising the SC and cerebellum integration (see Methods for a further description of the control loop and motor benchmark). Fig 1C displays the musculoskeletal upper limb model.

The following sections present the validation of the spino-cerebellar model with the lab recorded kinematics and EMG data; an evaluation of the SC effect in cerebellar motor adaptation in joint, synaptic and muscle spaces; and testing the response against external motor perturbations.

## 2.1 Spino-cerebellar and cerebellar models perform the recorded kinematics

We extracted kinematics and EMG data from two healthy participants (P1 and P2) performing upper limb movements in the vertical plane involving the shoulder and elbow (see Methods). The motor tasks performed by P1 and P2 can be grouped in: i) flexion-extension movements, ii) hand-tracked circular trajectories. Both motor task groups were performed at different

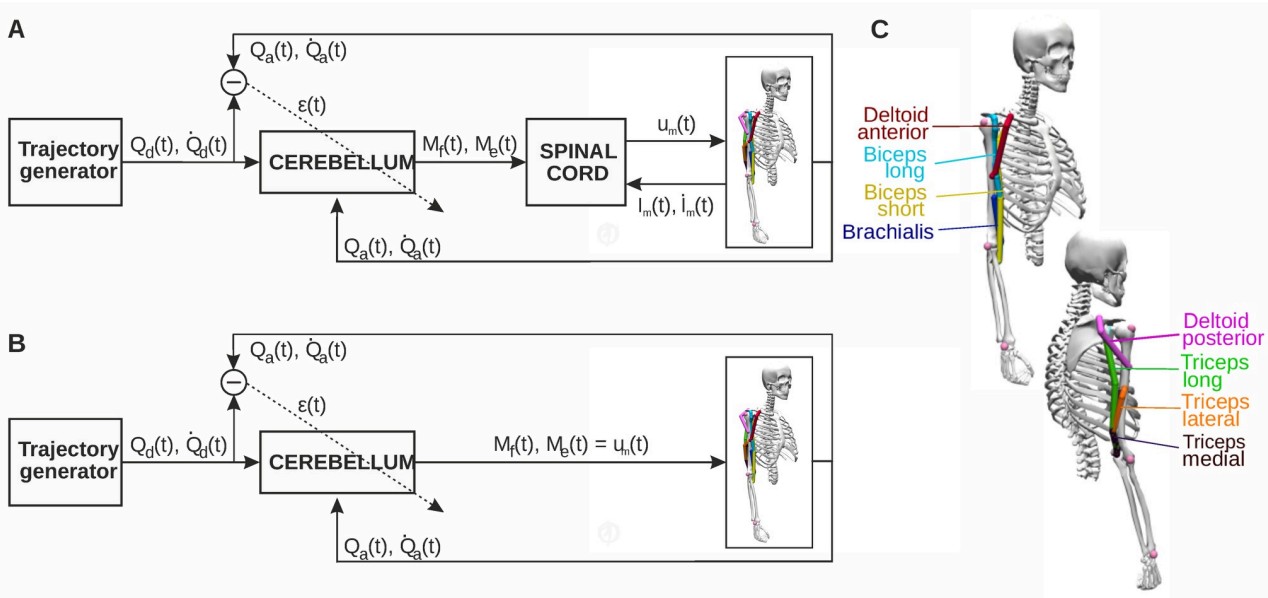

**Fig 1. Spino-cerebellar and cerebellar control loops. A)** Spino-cerebellar model. The cerebellum received the following input sensory information: the desired trajectory (joint position, $Q_d$, and velocity, $\dot{Q}_d$) coming from a trajectory generator, representing the motor cortex and other motor areas performing motor planning and inverse kinematics; the actual upper limb state (joint position, $Q_a$, and velocity, $\dot{Q}_a$) received from the musculoskeletal model; the instructive signal ($\epsilon$) obtained as the mismatch between the desired and actual joint state. The cerebellum then generated two output control signals per joint ($M_f$ and $M_e$, for flexor and extensor muscles, respectively), which were processed at the spinal cord. The spinal cord also received the muscle state (length, $l_m$, and velocity, $\dot{l}_m$) and generated the final muscle excitation signals ($u_m$) which actuated the musculoskeletal model. **B)** Cerebellar model. $M_f$ and $M_e$ were directly applied as muscle excitation signals commanded to the upper limb. For bi-articular muscles (biceps long and triceps long), the resulting $u_m$ was the mean of the control signal ($M_f$ or $M_e$) from both joints. **C)** Musculoskeletal model. The upper limb model adapted from Saul et al., 2014 [41] comprised two joints, shoulder and elbow, which were actuated by eight muscles: deltoid anterior and biceps long as shoulder flexors; deltoid posterior and triceps long as shoulder extensors; biceps long, short and brachialis as elbow flexors; triceps long, lateral and medial as elbow extensors. The images are extracted from OpenSim open source software (Seth et al., 2018 [42]).

speeds, thus providing a set of natural upper limb trajectories which constituted our initial motor control benchmark. We used the joint kinematics (i.e., shoulder and elbow position, $Q_d$, and velocity, $\dot{Q}_d$) extracted from the recording sessions as the desired trajectory to be learnt by the spino-cerebellar (Fig 1A) and cerebellar (Fig 1B) models in the simulation framework. Both models performed 3 repetitions of the motor adaptation process for each desired trajectory, each repetition consisting of 2000 consecutive trials, a trial-and-error process that allowed motor adaptation to fully deploy from scratch. The performance metric was given by the position and velocity mean absolute error (MAE), i.e., difference between the desired and actual trajectory in joint space, allowing to assess motor behaviour (see Methods).

We first calculated the position and velocity MAE evolution for both the spino-cerebellar and cerebellar models performing the trajectories extracted from each participant (Fig 2A, P1's 1.8s circle trajectory; Fig 2B, P2's 1.2s flexion-extension); see Supporting Information for all P1 and P2 motor tasks MAE evolution (S1(A)–S9(A) Figs). As the trajectory was repeated over time, the cerebellar adaptation allowed position and velocity error reduction. At the end of the motor adaptation process, both the spino-cerebellar and cerebellar models followed the target kinematics (Fig 2A and 2B); see Supporting Information for all P1 and P2 motor tasks kinematics performance (S1(B)–S9(B) and S1(C)–S9(C) Figs).

We found that, attending to the MAE mean and standard deviation (std) of the last 200 trials of the motor adaptation process (Fig 2C and 2D), the spino-cerebellar model reached better performance in terms of position tracking for all trajectories except for P1's slow (2.3s) and moderate (1.8s) flexion-extension (average position MAE for all motor tasks was 0.026 ± 0.011 rad for the spino-cerebellar model, and 0.037 ± 0.024 rad for the cerebellar model). Conversely, the cerebellar model reached better performance in terms of velocity tracking except for P1's slow (1.8s) circle and P2's moderate (1.6s) and fast (1.2s) circle (average velocity MAE for all motor tasks was 0.28 ± 0.07 rad/s for the spino-cerebellar model, and 0.26 ± 0.09 rad for the cerebellar model).

## 2.2 The spinal cord improves cerebellar learning convergence and speed

Once we revealed the adaptation capability of both the spino-cerebellar and cerebellar models, we studied the influence of the SC model on cerebellar learning over the adaptation process. Using the position and velocity MAE evolution of each P1 and P2 trajectory, we compared the spino-cerebellar and cerebellar models learning convergence and learning speed. To study learning convergence we applied control charts on the MAE data to determine the number of trials required to achieve a stable performance [43]. We computed the MAE mean ($\mu$) and standard deviation ($\sigma$) using a temporal sliding window with a sample size of 200 trials and defined different MAE limits relating $\mu$ and $\sigma$ (e.g., limit 1 = MAE $\in [\mu - \sigma, \mu + \sigma]$). We then measured the percentage of trials with a MAE value within each limit (see Methods). To check learning speed we analysed the number of trials required for the mean MAE of 200 samples to reach a given target (i.e., 0.1 rad for position MAE, and 0.5 rad/s for velocity MAE). Learning convergence and speed were tested on both position and velocity tracking performance (see Fig 3A for an example of position and velocity MAE evolution and the metrics used, see Methods for a further description).

The SC was proven to facilitate cerebellar learning as it reduced learning convergence time (Fig 3B), and increased learning speed (Fig 3C) for both position and velocity for both P1 and P2 trajectories (Fig 3 left and right column, respectively). Thus, cerebellar motor adaptation was shown to be: i) stabilised by the SC: average convergence time for $MAE_{pos}$ was 924 ± 423 trials for the spino-cerebellar model, and 1625 ± 526 trials for the cerebellar model; and for $MAE_{vel}$ 1015 ± 509 trials, and 1332 ± 571 trials, respectively; ii) accelerated by the SC: average

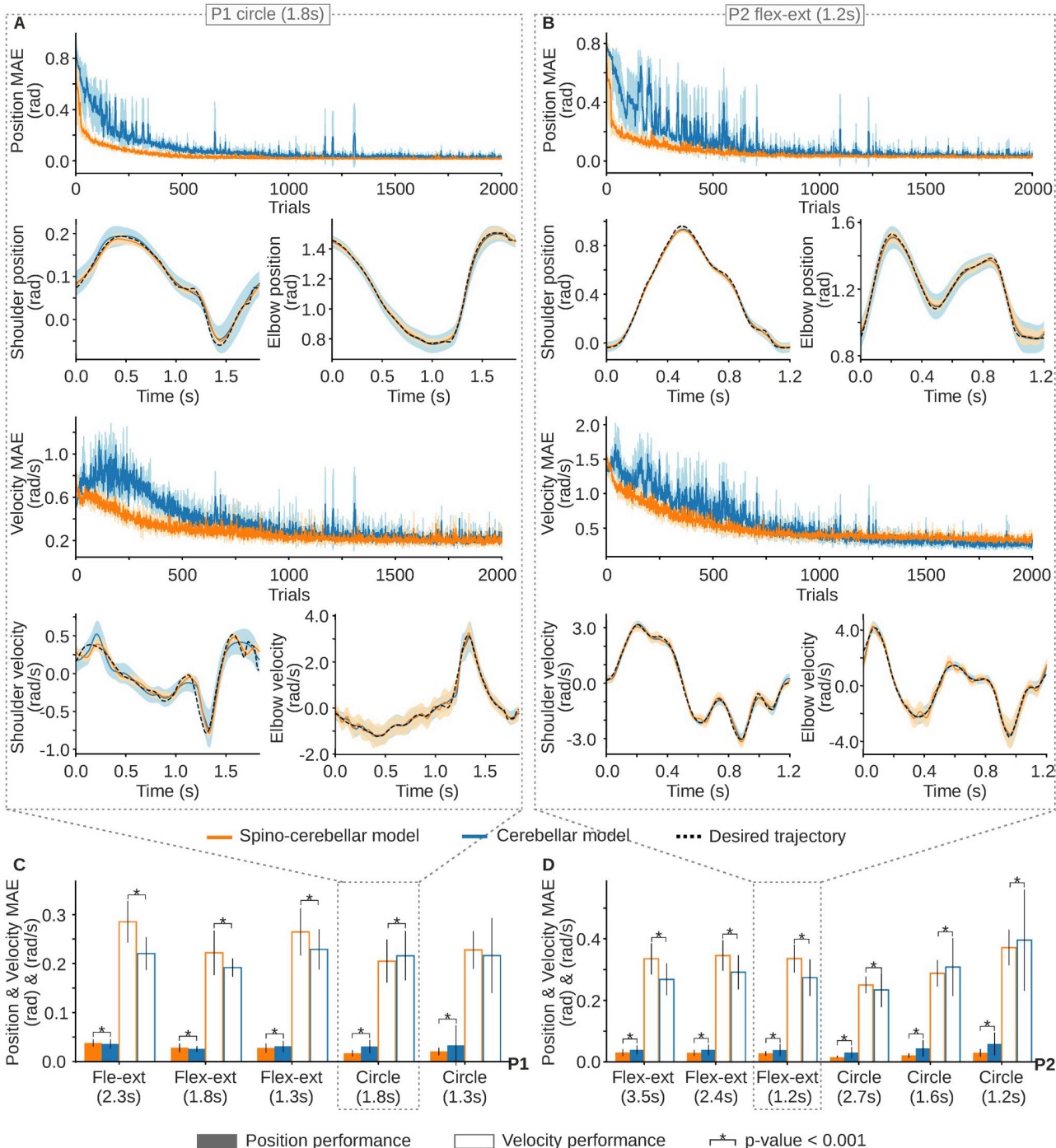

**Fig 2. Spino-cerebellar and cerebellar models kinematic performance for the lab recorded scenario. A)** Position and velocity mean absolute error (MAE) over the 3 repetitions of the 2000-trial motor adaptation process; and joint kinematics of the 3 repetitions last 200 trials (mean and standard deviation) for both the spino-cerebellar and cerebellar models performing P1's slow circle trajectory (1.8s), and **B)** P2's fast flexion-extension (1.2s). **C)** Mean and standard deviation of the position and velocity MAE (last 200 trials of the 3 trajectory repetitions) for all P1 recorded trajectories, and **D)** for all P2 recorded trajectories.

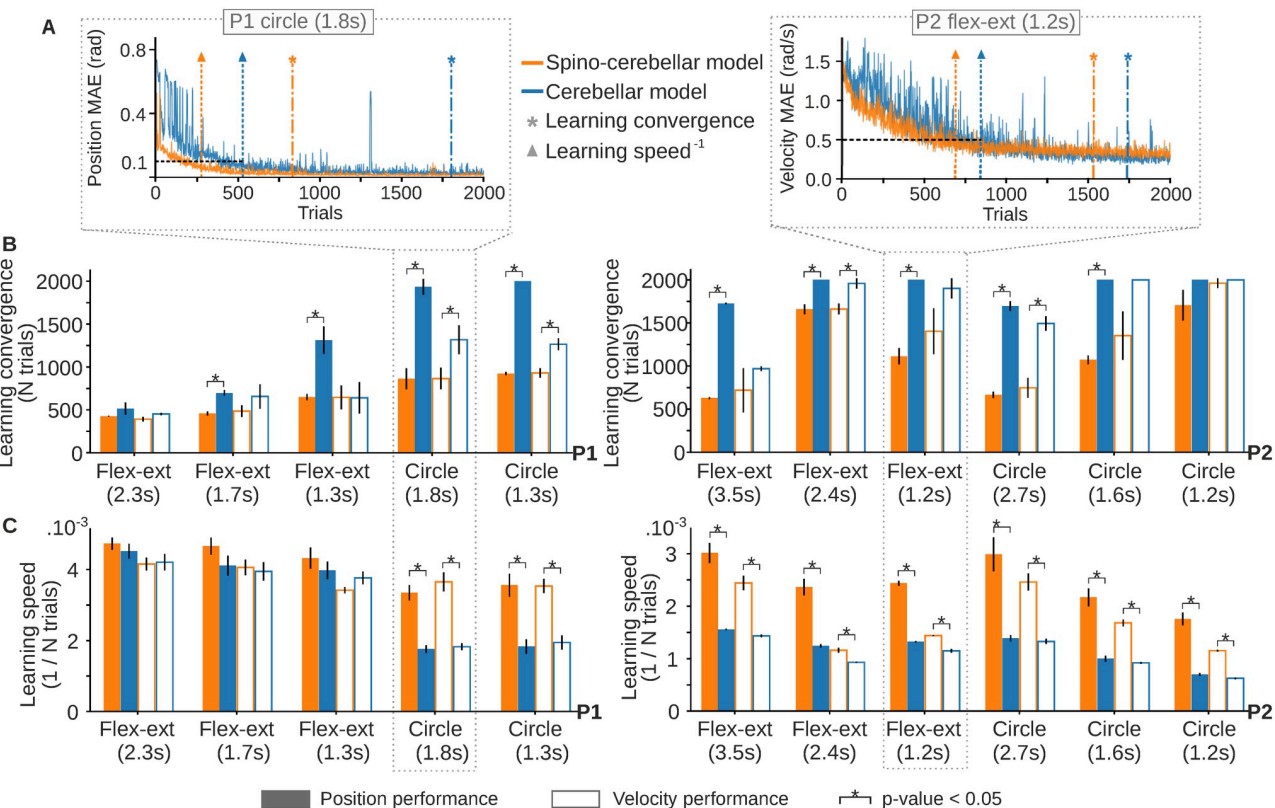

**Fig 3. Spino-cerebellar and cerebellar models motor adaptation for all P1 and P2 recorded trajectories. A)** Position MAE for the spino-cerebellar and cerebellar models for P1 slow circle (left column), and velocity MAE for both models performing P2 fast flexion-extension (right column). Both MAE plots show the trials at which the learning convergence and learning speed metrics are fulfilled. Only one repetition of the motor adaptation process is displayed. **B)** Learning convergence for both models and all trajectories from P1 (left column) and P2 (right column). The bar plots display the number of trials required by each model to fulfill the learning convergence criteria (see Methods). **C)** Learning speed for both models and all trajectories from P1 (left column) and P2 (right column). The bar plots depict the inverse of the number of trials required to reach a position MAE of 0.1 rad and a velocity MAE of 0.5 rad/s.

learning speed for position was $3.2e{-}3 \pm 1.0e{-}3$ trials$^{-1}$ for the spino-cerebellar model, and $2.1e{-}3 \pm 1.3e{-}3$ trials$^{-1}$ for the cerebellar model; and for velocity $2.7e{-}3 \pm 1.1e{-}3$ trials$^{-1}$, and $2.0e{-}3 \pm 1.3e{-}3$ trials$^{-1}$, respectively.

## 2.3 The spinal cord simplifies cerebellar synaptic adaptation at GC-PC

Consistently with the Marr-Albus-Ito cerebellar theory, learning in the cerebellum was provided by means of an STDP mechanism adjusting the synaptic weights at GC to PC synapses (a connection established through PFs, i.e., GC axons). The effect of the SC on cerebellar learning, already checked in terms of motor performance in the previous section, must leave its trace at the level of cerebellar synaptic adaptation. To conduct a direct comparison between the synaptic adaptation of the spino-cerebellar and cerebellar models, it is necessary to establish a common synaptic foundation. Thus, in one of the three repetitions of the 2000-trial motor adaptation process for each motor task, the synaptic weights between GCs and PCs were homogeneously initialised; i.e., at trial 0 all GC-PC synapses in both the spino-cerebellar and cerebellar models started with the same synaptic weight (4.8 nS). This homogeneously-initialised, 2000-trial repetition for the different P1 and P2 motor tasks, served as our

benchmark for synaptic adaptation. The common starting point concerning the synaptic weight distribution allowed for a fair comparison of the synaptic evolution between the two models.

During the motor adaptation process of both the spino-cerebellar and cerebellar models, we recorded the synaptic weight evolution at GC-PC connections every 200 trials for all P1 and P2 trajectories (Fig 4A and 4B). We then measured the entropy of the GC-PC synaptic weight distributions to quantify the synaptic complexity of both models: the higher the entropy, the more complex the synaptic weight distribution, i.e., higher heterogeneity of synaptic weights at the GC-PC population. The synaptic entropy metric can be grasped as measuring the complexity of the synaptic patterns displayed in Fig 4A and 4B, and the rest of the patterns recorded every 200 trials describing the full motor adaptation process. Contrasting the synaptic entropy of both models allowed evaluating the effect of the SC on cerebellar synaptic adaptation (Fig 4C and 4D). Noteworthy, results showed that for all motor tasks the SC reduced the entropy of the synaptic weight distribution: the mean entropy over all P1 and P2 trajectories was 3.65 ± 0.78 for the spino-cerebellar model, and 4.41 ± 1.14 for the cerebellar model. When the SC was lacking in the control loop, more complex synaptic patterns (i.e., higher specialisation) were required at cerebellar GC-PC connections. The spino-cerebellar model showed a simpler distribution of synaptic weights at GC-PC connections; in other words, the spinal cord was therefore shown to simplify learning in the cerebellum.

To deepen in the finding of the SC simplification of the cerebellar synaptic solutions, we analysed the amount of GC neurons required by the spino-cerebellar and cerebellar models. We measured the percentage of GC-PC synapses that experienced a modification of their initial weight as motor adaptation progressed, thus providing a measurement of how many GC-PC connections were effectively involved in motor learning (Fig 4E and 4F). Results showed that the spino-cerebellar model made use of fewer GC-PC connections for all P1 and P2 motor tasks. Thus, the SC allowed for a reduction of the GC neurons required for accurate execution of the motor tasks. The common synaptic starting point together with the synaptic entropy evolution and the amount of neurons involved in the motor adaptation process, support the divergence in the synaptic solutions acquired by the spino-cerebellar and cerebellar models, and the SC influence in facilitating cerebellar learning at the synaptic level.

## 2.4 Spino-cerebellar and cerebellar outcome in muscle space

We then evaluated the outcome in muscle space of both the spino-cerebellar and cerebellar models. We compared the recorded EMG envelopes to the main activated muscles from the spino-cerebellar and cerebellar models during performance of P1 and P2 trajectories. Fig 5 displays all the participants' recorded EMG signals and corresponding joint cocontraction index (CCI). Fig 6A illustrates the deltoid posterior (DELTpost) and brachialis (BRA) muscles during P1 slow circle performed by both the spino-cerebellar and cerebellar models, whilst deltoid anterior (DELTant) and triceps lateral (TRIlat) muscles are depicted for P2 fast flexion-extension. Both models reproduced the main activation patterns of each muscle with a small shift for P2 DELTant and TRIlat. The correlation between the spino-cerebellar or cerebellar activation and the EMG signals was generally larger than 0.5 (see Supporting Information (S13 and S14 Figs)). The correlation was, however, larger for the spino-cerebellar model for most of the muscles and scenarios. Nevertheless, the correlation averaged over muscles was similar between the two models for all the movements and we could not conclude on a better muscle pattern reproduction by one or the other model.

Results might not be conclusive when referred to a direct, muscle by muscle comparison between our models performance and the recorded EMG; note that our musculoskeletal upper

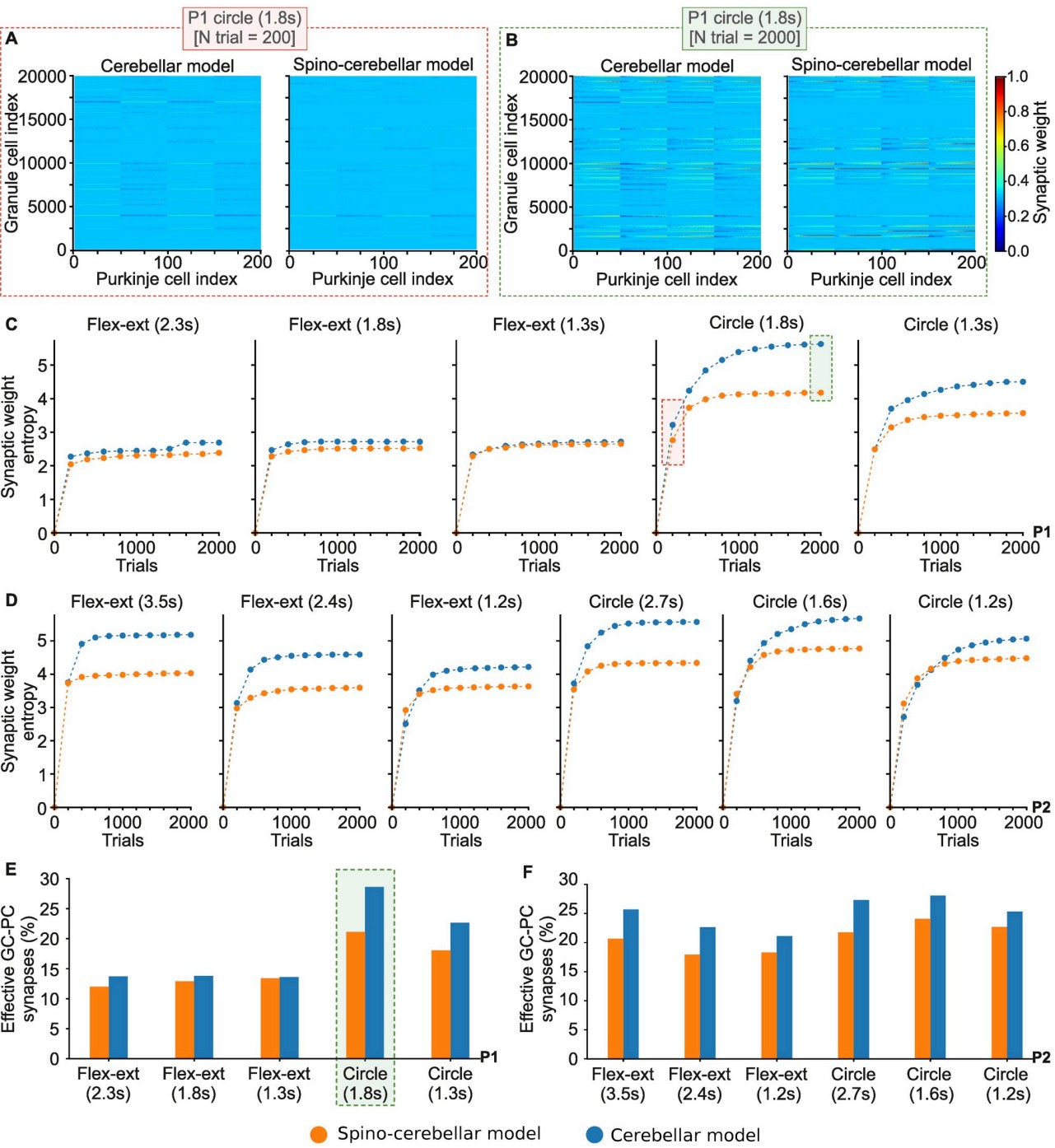

**Fig 4. Spino-cerebellar and cerebellar synaptic adaptation. A), B)** Synaptic weights at granule cell—Purkinje cell (GC-PC) synapses after 200 and 2000 trials, respectively, for both models performing P1's 1.8s circle trajectory. The heat map represents the normalised GC-PC synaptic weights, which could range from 0.0 to 15.0 nS. **C), D)** Evolution of the synaptic entropy at the GC-PC synapses over the 2000-trial motor adaptation process, for all P1 and P2 trajectories, respectively. The higher the entropy, the more complex the GC-PC synaptic distribution (i.e., higher heterogeneity in the synaptic weights of the GC-PC synapses). **E), F)** Percentage of GC-PC synapses that experienced a modification of their initial weight as motor adaptation progressed for all P1 and P2 trajectories; i.e., amount of GC-PC synapses required by both the spino-cerebellar and cerebellar models to succeed in the motor adaptation process.

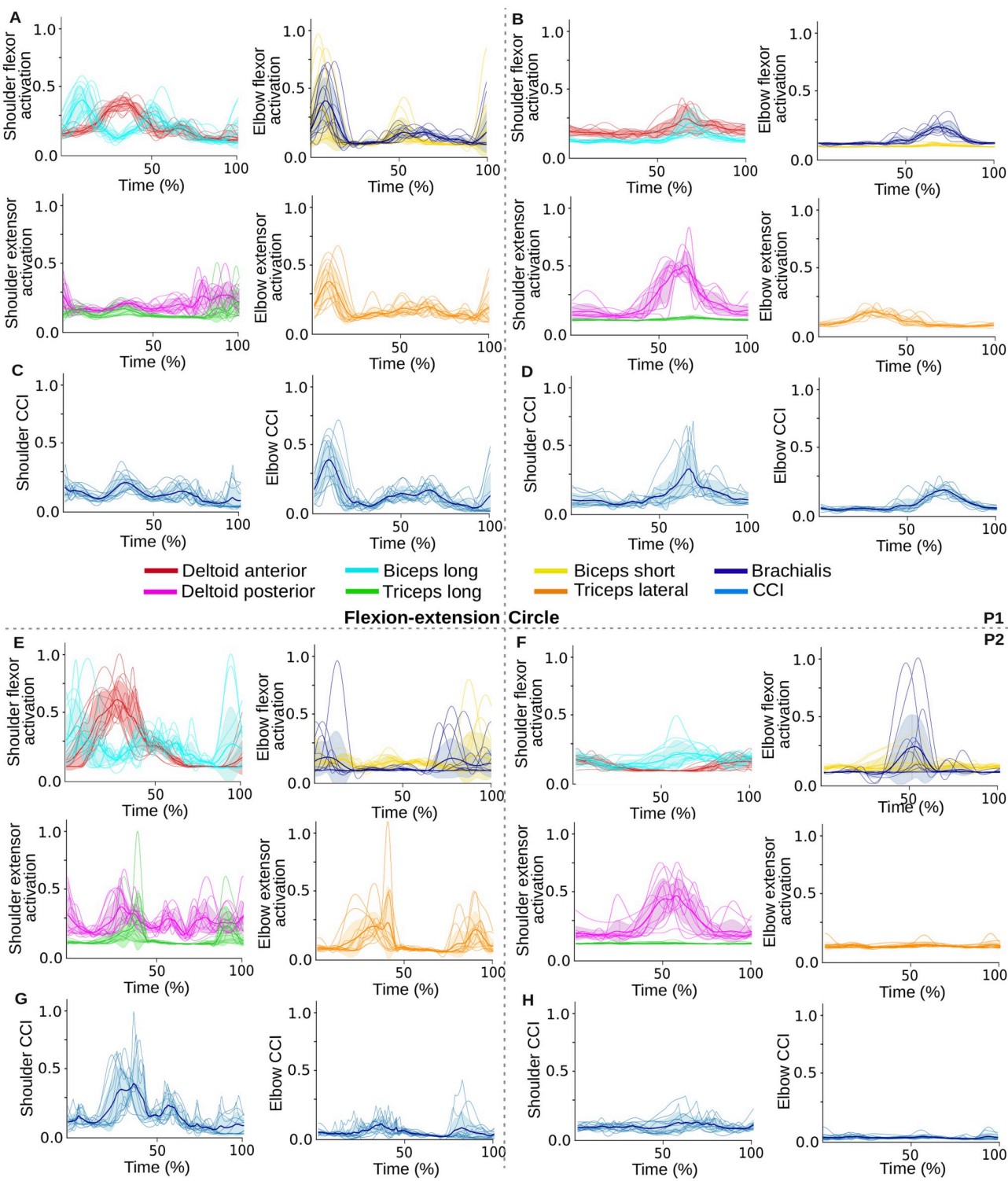

**Fig 5. EMG recordings obtained from both P1 and P2 during flexion-extension and circular movements. A)** EMG recordings obtained from P1 during flexion-extension movements and **B)** circular movements. The EMG signals of all recorded movements performed at different speeds are interpolated and displayed together to highlight the main activation patterns. Muscles are grouped together by joint (left column for shoulder, right column for elbow) and direction of motion (top row for flexor muscles, bottom row for extensor muscles). For simplicity, biarticular muscles (i.e., biceps long and triceps long) are only displayed on the shoulder group (left column). **C), D)** Resulting joint cocontraction index (CCI) from P1 during flexion-extension and circular movements, respectively. **E)** EMG recordings obtained from P2 during flexion-extension movements and **F)** circular movements. **G), H)** Resulting joint CCI from P2 during flexion-extension and circular movements, respectively.

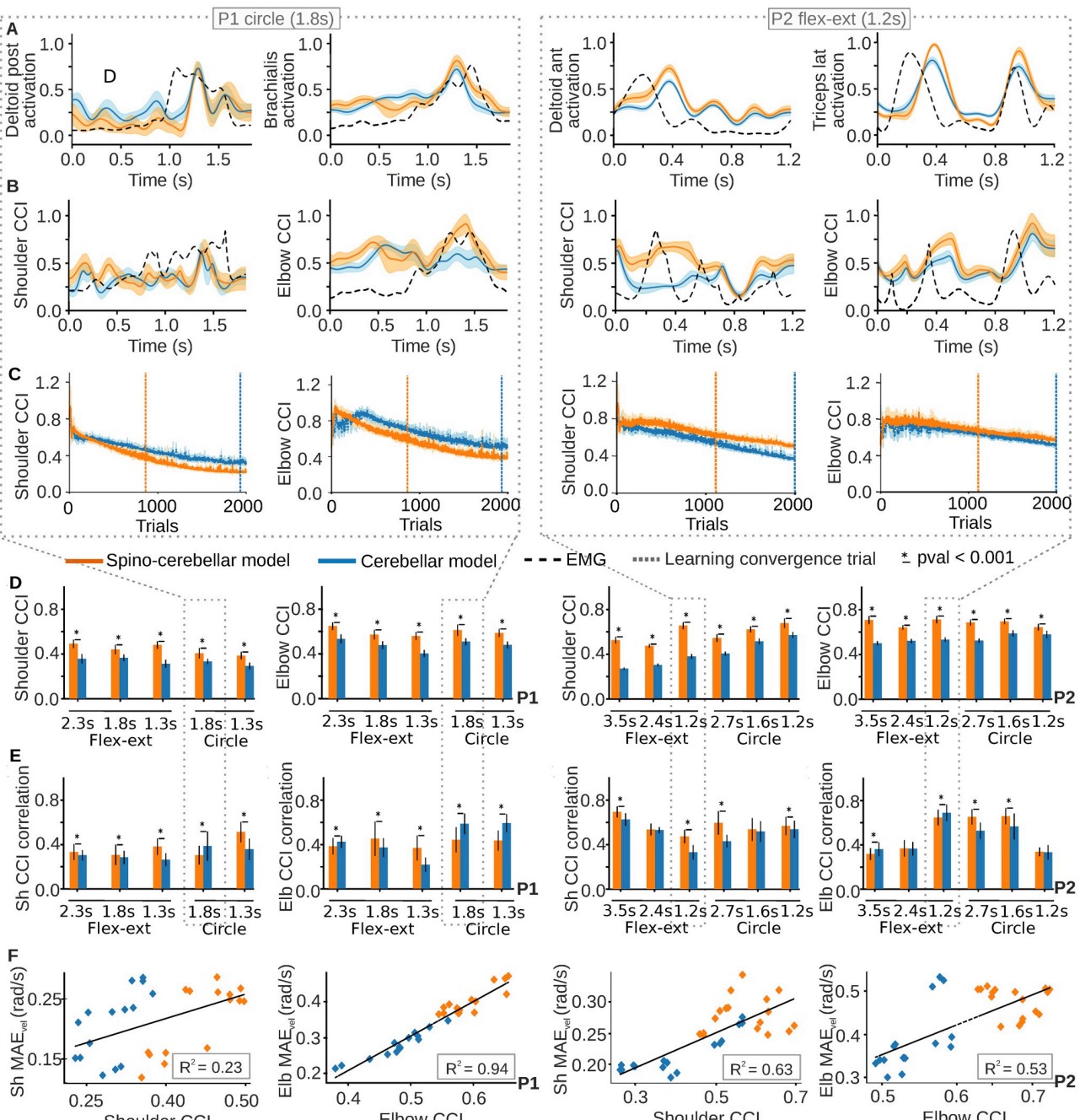

**Fig 6. Spino-cerebellar and cerebellar model performance in muscle space for all P1 and P2 recorded trajectories. A)** Comparison of muscle activation signals with recorded EMGs: the comparison only shows the main activated muscles during recordings of P1's slow circle (two left columns) and P2's fast flexion-extension (two right columns). The plots show the muscle activity of the 200 trials reaching the learning convergence metric, as well as their mean and standard deviation (std), for the two models performance. EMG signals are scaled by the maximum of the activation signals for each muscle for the sake of representation. **B)** Joint cocontraction index (CCI) from EMG activity and both models performance, for the trajectories represented in A). EMG CCI are scaled by the maximum of the models CCI for the sake of representation. **C)** Joint CCI evolution over the 2000-trial motor adaptation process. **D)** Joint CCI values for both models and all P1 (two left columns) and P2 (two right columns) trajectories. **E)** Joint CCI correlation between the models and EMG for all P1 (two left columns) and P2 (two right columns) trajectories. **F)** CCI-$MAE_{vel}$ relation: linear regression between joint CCI and joint $MAE_{vel}$ over all the trajectories from P1 (two left columns) and P2 (two right columns).

limb model was actuated by eight muscles, a mere simplification of the complex muscle dynamics of the human upper limb. To overcome this, we further studied performance in muscle space using the joint cocontraction index (CCI), which unifies muscle activity per joint and provides a more comprehensive analysis (refer to the Methods section for details on the CCI computation procedure).

Both the spino-cerebellar and cerebellar model exhibited a gradual decrease in joint CCI as motor adaptation evolved (see S15 fig for CCI evolution). Due to computational constraints, each trajectory repetition was limited to 2000 trials, at the end of which CCI still showed a decreasing trend. The spino-cerebellar model displayed higher CCI values than the cerebellar model in some cases, whilst lower CCI values in others. Consequently, we could not draw definitive conclusions on the final CCI values. Instead, since we used convergence of the kinematic performance as our learning metric, we focused on the CCI values as the kinematic adaptation progressed.

During the early stages of learning, the spino-cerebellar model exhibited higher overall CCI values than the cerebellar model (see Table 1). Subsequently, we measured the joint CCI once an accurate internal model was developed, as indicated by a stable kinematic performance (see Methods for details on learning convergence metrics). We found that the spino-cerebellar model better reproduced the CCI patterns at the level of the elbow for P1 slow circle and at the level of the shoulder for P2 fast flexion-extension (Fig 6B). Significantly, the spino-cerebellar model provided higher CCI values at the kinematic convergence point (Fig 6C) for all P1 and P2 trajectories, both for the shoulder and elbow (Fig 6D). We then compared the CCI provided by both models with the CCI from the recorded EMG (Fig 6E). The correlation was mainly higher for the spino-cerebellar model. We observed a similar trend as that observed for

**Table 1. Joint CCI mean and standard deviation values during the first 400 trials for each P1 and P2 trajectory, for both the spino-cerebellar and cerebellar model.**

| Trajectory | Joint | Spino-Cerebellar CCI | Cerebellar CCI | p-value $<0.005$ |
|---|---|---|---|---|
| P1 flex-ext slow | Shoulder | $0.58 \pm 0.12$ | $0.52 \pm 0.13$ | * |
| | Elbow | $0.75 \pm 0.07$ | $0.69 \pm 0.10$ | * |
| P1 flex-ext mod | Shoulder | $0.56 \pm 0.12$ | $0.56 \pm 0.12$ | |
| | Elbow | $0.71 \pm 0.08$ | $0.70 \pm 0.10$ | * |
| P1 flex-ext fast | Shoulder | $0.61 \pm 0.10$ | $0.60 \pm 0.10$ | |
| | Elbow | $0.72 \pm 0.06$ | $0.70 \pm 0.08$ | * |
| P1 circle slow | Shoulder | $0.60 \pm 0.09$ | $0.60 \pm 0.07$ | |
| | Elbow | $0.82 \pm 0.08$ | $0.81 \pm 0.09$ | * |
| P1 circle mod | Shoulder | $0.54 \pm 0.09$ | $0.59 \pm 0.07$ | * |
| | Elbow | $0.78 \pm 0.06$ | $0.77 \pm 0.06$ | * |
| P2 flex-ext slow | Shoulder | $0.69 \pm 0.10$ | $0.64 \pm 0.07$ | * |
| | Elbow | $0.81 \pm 0.07$ | $0.78 \pm 0.08$ | * |
| P2 flex-ext mod | Shoulder | $0.73 \pm 0.09$ | $0.66 \pm 0.06$ | * |
| | Elbow | $0.83 \pm 0.06$ | $0.78 \pm 0.07$ | * |
| P2 flex-ext fast | Shoulder | $0.76 \pm 0.07$ | $0.71 \pm 0.05$ | * |
| | Elbow | $0.77 \pm 0.07$ | $0.72 \pm 0.07$ | * |
| P2 circle slow | Shoulder | $0.68 \pm 0.07$ | $0.68 \pm 0.05$ | |
| | Elbow | $0.73 \pm 0.07$ | $0.73 \pm 0.07$ | |
| P2 circle mod | Shoulder | $0.78 \pm 0.06$ | $0.71 \pm 0.05$ | * |
| | Elbow | $0.74 \pm 0.09$ | $0.70 \pm 0.09$ | * |
| P2 circle fast | Shoulder | $0.86 \pm 0.07$ | $0.70 \pm 0.04$ | * |
| | Elbow | $0.68 \pm 0.08$ | $0.65 \pm 0.07$ | * |

$MAE_{vel}$, therefore, we performed a linear regression between the CCI and $MAE_{vel}$ for each joint. The results (Fig 6F) highlighted a linear trend between these quantities for P1 elbow and P2 shoulder and elbow (with a coefficient of determination of 0.94, 0.63 and 0.53 respectively), whereas P1 shoulder presented a weaker relationship (with a coefficient of determination of 0.23).

Overall, we highlighted various findings that were consistent for various trajectories with various initial and final positions and speeds. The spino-cerebellar model provided more stable and faster learning with simpler cerebellar synaptic adaptation. Furthermore both models exhibited a gradual reduction in cocontraction as learning progressed; however, the spino-cerebellar model reached learning convergence with higher CCI values and better correlation to the recorded EMG.

## 2.5 The spinal cord increases the robustness against motor perturbations

The experimental setup used in the previous sections for the subjects recording sessions did not provide neural activity data, thus preventing any conclusions about the roles of the cerebellum and SC in response to external perturbations. It was not possible to determine whether the cerebellum or SC would be the main contributor to the subjects' response to external perturbations. To circumvent this data limitation, we used our computational approach. By including or removing the SC from the control loop we were able to investigate the SC influence in responding to perturbations. To study the response against external perturbations of both the spino-cerebellar and cerebellar models, we used our lab designed benchmark: upper limb flexion-extension movements with bell-shaped velocity profiles, characteristic of reaching movements [44]. This kind of movement is usually used for addressing active-limb control malfunctioning, as cerebellar patients usually display upper limb oscillatory tremors that result in endpoint overshooting and undershooting when reaching a target [45].

Both models faced 2000 consecutive trials of the flexion-extension movement performed at different speeds (3s, 2.3s, 1.5s); after motor adaptation, both models succeeded in performing the target kinematics (see Supporting Information, S16 fig). Once both models adapted to perform the desired trajectories, we tested the contribution of the SC in handling motor perturbations. For that, we induced a set of external forces: i.e., 50 N for 30 ms applied to the hand in different directions and at different points along the flexion-extension movement, resulting in kinematic deviation (Fig 7A). We then measured the MAE deviation from the ideal, no-perturbation scenario (Fig 7B–7E). Each perturbation type was applied on 50 separate trajectory trials to get an average response (see Methods). Besides, to gain a deeper understanding of the SC involvement in handling perturbations, we individually assessed the influence of the stretch reflex and reciprocal inhibition mechanisms. To that end, we applied the aforementioned set of external perturbations to two additional cases: i) SC equipped with just stretch reflex (SR-cerebellar model); ii) SC equipped with just reciprocal inhibition (RI-cerebellar model). In order to focus on the contribution of the two SC mechanisms and conduct a direct comparison between them, for the SR-cerebellar and RI-cerebellar cases, we set the cerebellar synaptic weights to those developed by the spino-cerebellar model. More specifically, we equipped the cerebellar network with the synaptic solution acquired by the cerebellum after motor adaptation when the SC model was fully equipped, thus preventing the cerebellum from learning how to compensate for the deficiencies induced by the lack of spinal mechanisms. The cerebellar learning capability was disabled in all four cases to prevent adaptation to the perturbations. Note that, in previous research, we studied the cerebellar ability to effectively handle external perturbations during motor adaptation under changing task conditions [18, 20]. By disabling the cerebellar learning capability, we could specifically focus on the SC contribution.

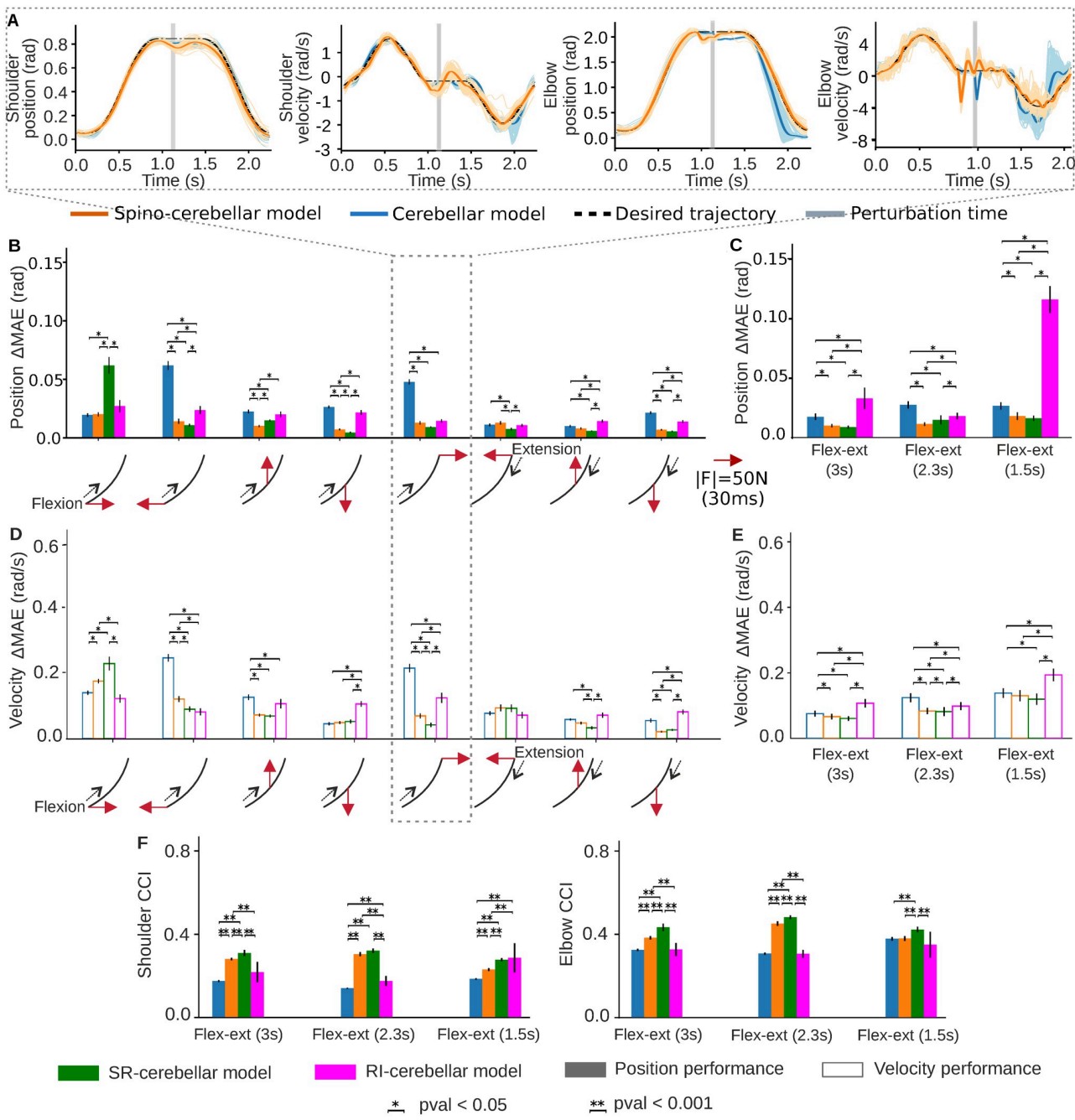

**Fig 7. Spino-cerebellar, SR-cerebellar, RI-cerebellar and cerebellar model responses to external motor perturbations during bell-shaped flexion-extension trajectories. A)** Kinematic performance of both the spino-cerebellar and cerebellar models under a forward perturbation at the flexed position whilst performing the 2.3s flexion-extension trajectory. 50 trials are displayed. **B)** Position deviation ($\Delta \overline{MAE}$) caused by all the perturbations applied during the 2.3s flexion-extension trajectory for the four models. Mean $\Delta \overline{MAE}$ and standard deviation (std) of 50 trials are displayed. **C)** Mean position $\Delta \overline{MAE}$ and std for all the perturbations applied to the flexion-extension trajectories performed at different speeds (3s, 2.3s, 1.5s). 50 perturbed trials were used for each perturbation type. **D)** Velocity deviation ($\Delta \overline{MAE}$) caused by all the perturbations applied during the 2.3s flexion-extension trajectory for the four models. Mean $\Delta \overline{MAE}$ and std of 50 trials are displayed. **E)** Mean velocity $\Delta \overline{MAE}$ and standard deviation for all the perturbations applied to the flexion-extension trajectories performed at different speeds (3s, 2.3s, 1.5s). 50 perturbed trials were used for each perturbation type. **F)** Shoulder and elbow CCI values for the four models. Mean and std of 50 trials without perturbation are displayed.

The kinematic performance of both the full spino-cerebellar and cerebellar models under one perturbation type, whilst performing the moderate flexion-extension movement (2.3s), shows that the cerebellar model exhibits a larger kinematic deviation compared to the spino-cerebellar model, particularly at the elbow level Fig 7A). When analysing the responses of the four models (spino-cerebellar, cerebellar, SR-cerebellar, and RI-cerebellar) to the set of perturbations applied during the moderate flexion-extension trajectory (Fig 7B and 7D), we found that: i) the spino-cerebellar model exhibited significantly smaller Mean Absolute Error deviation ($\Delta MAE$) compared to the cerebellar model, both in terms of position and velocity, for the majority of the applied perturbations; ii) the SR-cerebellar model exhibited smaller $\Delta MAE$ compared to the RI-cerebellar model. Additionally, the SR-cerebellar and spino-cerebellar models exhibited similar deviations for most of the perturbations. Similar results were obtained for the slow and fast bell-shaped flexion-extension trajectories (please refer to Supporting Information for the corresponding figure, S17 fig).

The kinematic performance analysis on the deviations of the four models in response to all the applied perturbations during the different trajectories (Fig 7C and 7E), confirmed the similar behaviour of the spino-cerebellar and SR-cerebellar models. Both models exhibited superior performance and greater robustness compared to the cerebellar and RI-cerebellar models. All these findings support the role of the SC in handling external motor perturbations, and suggest that the stretch reflex component plays a dominant and more effective role in dealing with perturbations, whilst the reciprocal inhibition mechanism is not as extensively involved.

Finally, we computed the joint CCI values of our four models (Fig 7F) during the three trajectories without perturbations. This analysis aimed to determine whether the CCI could serve as a biological marker to predict the performance against perturbations. The results showed that the SR-cerebellar model exhibited the highest CCI values, followed by the spino-cerebellar model, the RI-cerebellar model, and lastly the cerebellar model. This observation indicates a correlation between higher CCI values and greater robustness against perturbations, as models with higher CCI values exhibited smaller trajectory deviations. Based on these findings, we can conclude that the presence of the spino-cerebellar pathway contributes to better handling external motor perturbations, with the stretch reflex playing a prominent role, which leads to increased cocontraction levels.

## 3 Discussion

The integration of biologically plausible computational models of neural regions allows studying their interaction and complementarity. We presented a computational exploratory approach integrating a cerebellar and an SC model, performing motor control of an upper limb musculoskeletal model; a simulation framework complemented with kinematic and EMG data validation. We contrasted the spino-cerebellar integrated model with a cerebellar model, both performing in the same motor benchmark, which allowed us to extract some key elements of the kinematic and muscle performance directly attributable to the presence of the SC in the spino-cerebellar control loop. The SC was found to stabilise, accelerate, and facilitate cerebellar motor adaptation and to improve the response against perturbations through stretch reflexes and reciprocal inhibition. Rather than being an evolutionary constraint, the SC offers motor control benefits.

Both the spino-cerebellar and cerebellar models succeeded in learning the musculoskeletal dynamics to achieve the goal motor behaviour. Noteworthy, the presence of the SC provided faster motor adaptation, thus assisting cerebellar learning, in line with previous findings on the SC circuitry facilitating motor control of musculoskeletal dynamics [46]. In this regard, a significant finding was the fact that the spino-cerebellar model revealed less complexity at the

GC-PC synaptic weight distribution and it required fewer GC neurons for accurate execution of the motor tasks: the SC led to the formation of less specialised GC-PC synapses, thus freeing up computational resources within the cerebellum. The cerebellar granular layer can be compared to a reservoir computing mechanism [47, 48], wherein the cerebellum increases the dimensionality of the sensorimotor inputs that it receives. The ability of the SC to facilitate a more efficient use of GC neurons (the most numerous neuron type in the mammalian brain [49]) can be compared to increasing the number of units in reservoir computing. Increasing the size of the reservoir enhances its computational power and expands its memory capacity [50, 51]. A larger size increases the degrees of freedom of the reservoir response, allowing it to capture more complex dynamics. A larger size also allows the reservoir to store greater amounts of values. Thus, by facilitating a more efficient use of cerebellar resources, the SC enables an increase in the computational capacity of the cerebellum, hence overcoming its physical limitations. To the best of our knowledge, it is the first time that a computational model highlights and weighs the influence of the SC in facilitating cerebellar learning.

Direct regulation of muscle activity by the SC has here been found to facilitate the cerebellar acquisition of the upper limb inverse dynamics. Indeed, the body plant dynamics to be learnt by higher brain areas, might be simplified by the SC taking over lower level and faster control primitives, such as the SC potential role in gravity compensation [52, 53]. Thus, the SC performance in muscle space may lighten other operations of the sensorimotor process, occurring at a higher level such as the cerebellum's contribution in compensating interaction torques in joint space [54], or in shaping spatiotemporal muscle synergies rather than generating specific complex muscle patterns [55]. High-order brain functions, such as learning generalisation, have been pointed as key aspects that enable the brain to overcome its physical limitations [22, 56]. Here we highlight the interaction between different CNS regions as pivotal for enhancing the brain computational capacity and overcoming its physical constraints.

There is biological evidence supporting higher cocontraction levels during the early stages of learning, e.g., infants exhibit higher cocontraction during stepping motions compared to adults, and cocontraction is subsequently reduced with practice [57]; in the case of the upper limbs, higher cocontraction levels resulting in higher joint stiffness during learning and adaptation to new dynamics have been reported [58, 59]. These findings suggest that higher cocontraction levels during early learning stages enhance the learning rate and facilitate the acquisition of internal models, which once acquired, enable a gradual reduction of cocontraction [60]. In our work, both the spino-cerebellar and the cerebellar model exhibited a gradual decrease in cocontraction levels as the motor adaptation process evolved; thus, both models displayed a biologically plausible behaviour in the broader context of reducing cocontraction through learning. Importantly, the spino-cerebellar model exhibited an overall higher cocontraction level during early stages of learning, and it also provided higher joint CCI values at the learning convergence point, i.e., when an accurate internal model of the upper limb was fully acquired.

The SC stabilises the system at muscle level, increasing cocontraction through stretch reflexes and coordinating the antagonist activation patterns through reciprocal inhibition. Thus, the SC participates in modulating cocontraction, which plays an important role in motor control and stability [38, 40], providing better accuracy despite its energy cost [39]. In our framework, the spino-cerebellar model increased the joint CCI whilst the upper limb internal model was being acquired; i.e., cocontraction was indeed mostly determined by the SC motor action. Importantly, the CCI from the spino-cerebellar model also resulted in a better correlation with the CCI patterns from the recorded EMG signals, thus supporting closer biological plausibility than the cerebellar model; incorporating more detailed biological motor control mechanisms into the model increases the level of physiological plausibility in the

results [46]. The CCI increment was also revealed when inducing perturbations in the control loop; the spino-cerebellar model provided a better response, reducing the kinematic deviation. Further analysis of the SC mechanisms revealed the stretch reflex to provide better responses to perturbations than the reciprocal inhibition mechanism (SR-cerebellar vs. RI-cerebellar models), showing smaller kinematic deviation whilst also exhibiting higher joint CCI values. Thus, the stretch reflex was found to be a dominant mechanism in the SC improvement of kinematic performance under external perturbations, whilst the inhibitory action of the reciprocal inhibition was not as extensively involved. Muscle elasticity has been previously pointed as a significant passive contributor in handling perturbations [61]. Within our framework, the spino-cerebellar and cerebellar models, as well as the SR-cerebellar and RI-cerebellar models, all used identical muscle models with the same mechanical properties. Consequently, the passive muscle contribution to handling perturbations remained equal in all cases. This direct equivalence allowed us to focus on the contribution of the SC and attribute a pivotal role to the SC through the stretch reflex in providing robustness against external perturbations, thus supporting previous physiological and modelling research [24, 25, 32, 33].

The cocontraction increase carried by the SC involved a poorer velocity tracking. Indeed, the spinal reflexes between antagonist muscles may induce oscillatory activation patterns and thus alter the velocity performance. We did not observe, however, any trend in CCI values related to movement speeds despite higher cocontraction values have been reported in slower movements [40]. Due to the SC and cerebellar models conception, our implementation lacks differentiation between the roles of the cerebellum and SC depending on movement speed. Note however that it is expected a major role of the cerebellum in fast ballistic movements which cannot rely on feedback availability [36, 62], and which do present lower cocontraction levels [40].

Our model could be further improved by adding other cocontraction mechanisms to the control loop. Clinical studies supported a potential role of the cerebellum and basal ganglia in cocontraction mechanisms. In particular, patients with cerebellar ataxia showed excessive agonist-antagonist coactivation [63] and cerebellar stimulation was shown to reduce coactivation in patients with spasticity [64]. Thus, future development of the cerebellar model shall include control of the cocontraction level. On the SC side other pathways could be included, in particular modulation mechanisms that are present during arm movements [25, 28, 29]. For instance, presynaptic inhibition of Ia terminals at both activated and antagonist pathways is slightly decreased at the onset of a voluntary contraction through descending signals. Thus, the increased gain of the stretch reflex pathway ensures that activated motoneurons receive Ia feedback support. The reciprocal Ia inhibition is also depressed during a voluntary contraction at the corresponding muscle to prevent its inhibition by the stretch-induced Ia discharge from its antagonist. During cocontraction, reciprocal Ia inhibition is also depressed by increased presynaptic inhibition on Ia terminals [5]. Also synaptic plasticity could be included in the SC model, as done in previous computational approaches [34]. Activity dependent plasticity mechanisms have been reported in the SC: e.g., the spinal stretch reflex can indeed be conditioned [65]; the feedforward circuits within the SC, in addition to somatosensory feedbacks, may contribute to SC learning by allowing motoneurons to contrast feedforward and feedback motor inputs [66]. Supporting the latter, [67] showed that signals in human muscle spindle afferents during unconstrained wrist and finger movements predict future kinematic states of their parent muscle. Muscle spindles would then have a forward-sensory-model role, as that attributed to the cerebellum [68], emphasising the complementarity and overlapping functionality between neural regions.

Integrated computational models represent a powerful tool to support and guide experimental studies in the pursuit of a better understanding of the CNS. We believe our spino-

cerebellar model to contribute in this direction, providing a picture of how the SC influences cerebellar motor adaptation and learning. Further development of the model, together with addition of other neural regions, will help to keep elucidating CNS operation.

## 3.1 Cerebellum and spinal cord modelling assumptions

The implemented cerebellar and spinal cord computational models are physiologically based and adhere to the principle of biological plausibility. Nonetheless, certain assumptions were made to facilitate interpretation of the results.

Regarding the cerebellum, it is well-known for its role in motor adaptation and learning, a role mainly supported by the cerebellar ability to acquire internal models of both body-plant dynamics and external objects [68]. These internal models are found to be either forward or inverse; a forward model maps the sensorimotor state to its predicted motor behaviour, whilst an inverse model maps the desired behaviour to the motor commands that will make it possible. The literature addresses the existence of both inverse and forward models in the cerebellum and, rather than confronting the two model alternatives, there is theoretical, computational, and behavioural experimentation that supports the coexistence and complementarity of both approaches [19, 68–71]. When modelled as a forward model, the cerebellum modulates the descending motor commands from the motor cortex to correct the mismatch between predicted and actual motor behaviour. In such cases, the body-plant dynamics are learnt at higher brain areas, and the cerebellum operates as a corrective mechanism in addition to the already learnt dynamics. Conversely, when functioning as an inverse model, the cerebellum does not rely on descending motor commands. Instead it can directly provide the entire motor output and bypass the motor cortex [72]. In our work, the cerebellum is implemented as an inverse model, therefore putting it in a worst-case scenario in which it needs to learn the entire body-plant dynamics. The purpose of this approach is to directly address the influence that the SC has on cerebellar motor learning.

If the implemented cerebellar model were a forward model, it might improve tracking performance; however, it would hinder the evaluation of the influence of the SC on cerebellar learning. The SC has a direct and fast action over muscles, thus leading to significant modifications of the arm-plant dynamics. The SC modification of the arm-plant dynamics would affect the cerebellar learning whilst also influencing the motor cortex output. If the SC control mechanisms were already being accounted for at higher brain areas, it would indeed pose challenges in evaluating and quantifying the SC effect on cerebellar learning. It would be difficult to compare the impact of the SC between our spino-cerebellar and cerebellar cases due to the potential overlapping factors incorporated by the already accounted SC control mechanisms at higher brain areas. Using the cerebellum as an inverse model provides a direct means to assess the effects of the SC on cerebellar motor learning. By isolating the role of the cerebellum in motor control, we were able to study how the SC influences the cerebellar motor learning, i.e., comparison between our spino-cerebellar and cerebellar cases is straightforward.

To further facilitate the comparison between our spino-cerebellar and cerebellar cases, we simplified the action of the cerebellar output commands, by omitting other inputs to the spinal cord, e.g. from the motor cortex and basal ganglia. In our cerebellar case, motor behaviour is solely driven by the cerebellum as the only nervous region present in the control loop. Therefore, in this case, the cerebellar output commands directly controlled muscle activation. In the spino-cerebellar case, we maintained the cerebellar descending commands to the SC as direct control signals, which acted upon motoneurons, thus ensuring equivalent cerebellar output commands in both the spino-cerebellar and cerebellar cases. This simplified descending pathway facilitated directly studying the influence of the SC on cerebellar motor learning. Notably,

our findings suggest that the SC facilitates cerebellar learning, as evidenced by the differences in the developed cerebellar solutions with and without SC involvement (i.e., spino-cerebellar vs. cerebellar cases exhibited different GC-PC synaptic weight distributions).

However, the intricate reality of the nervous system presents greater complexity; a significant convergence of of cerebral and cerebellar efferent pathways occurs within the SC interneuronal circuitry before reaching motoneurons [73]. This raises a key question: do cerebellar internal models include this intermediary circuitry and account for the interaction amongst descending pathways, beyond the dynamics of the musculoskeletal system?

Our work revealed that cerebellar learning is impacted by the interaction with SC circuitry, and therefore that spinal cord properties should not be underestimated when studying learning. Other descending pathways, such as those from the pyramidal cortical system likely also impact cerebellar learning. Given that biological motor learning is determined by the interplay amongst various nervous regions, future research should expand this study to explore neural interactions that contribute to this intricate process.

Regarding the cerebellar sensory input, it has been previously stated that within the dorsal spino-cerebellar tract (DSCT) there is a subpopulation of neurons representing whole limb parameters and an equally-sized subpopulation representing joint-related signals. This halved structure allows for an early processing stage of sensory information that facilitates a reference frame for whole limb kinematics [74]. In our work, the focus was not on this early sensory processing in the DSCT. Instead, the proprioception signals provided by the SC in joint space were directly sent to the cerebellum. Nonetheless, despite its joint-based architecture, the cerebellum did account for whole limb kinematics as the driving force behind the motor adaptation process.

Our cerebellar model builds upon Ito's inverse model assumption [72], which derived from oculomotor system studies. Here, we extend that cerebellar scheme to the upper limb musculoskeletal system, by including a subset of multi-articular muscles, with their associated dynamics, proprioceptive afferents and SC circuits. The structure of our cerebellar model involves assigning a cerebellar microcomplex to each joint, following Ito's microcomplex theory [72]. Whilst the input sensory signals and output motor signals are joint-based, i.e., divided in a modular organisation of different cerebellar functional units [75], the cerebellar learning at GC − PC synapses, in contrast, considers the entire limb kinematics. GCs are believed to perform a recoding of sensory inputs [14, 76], ensuring an univocal and unambiguous representation of the kinematic state of the limb. GCs, in turn, transmit this sensory state to PCs which, driven by the CFs instructive action, adapt their activity to cope with the desired motor behaviour. Importantly, PCs are innervated by a massive number of GCs, with an estimated $\sim 175{,}000$ GC-PC synapses in the rat cerebellar cortex for each PC [77]. Amongst that massive GC-PC innervation, PCs receive excitatory inputs not only from GCs within the very same microcomplex but also from GCs belonging to other microcomplexes. This connectivity allows the linkage of different cerebellar microcomplexes through GC-PC synapses [75, 78]. In our model, each PC receives excitatory input from all GCs, not just from those associated with the very same microcomplex. Therefore, each PC receives the entire sensory information from GCs, and a specific teaching/instructive signal through a particular CF corresponding to the same microcomplex, in accordance to CF-PC one-to-one synapses [79]. This connection arrangement makes synaptic plasticity at PCs to be affected by the sensory information from all joints and enables the cerebellum to acquire a comprehensive internal model representation of the upper-limb, rather than a joint-specific representation. Therefore, despite the microcomplex structure being based on individual joints, the ensemble activity of GCs across PCs allows for motor adaptation to account for the entire limb kinematics.

Regarding modelling of the SC circuits, some assumptions were also made. The synaptic weights of the spinal pathways were selected aiming to reproduce physiological connectivity by maintaining the ratio between the amplitudes of excitatory postsynaptic potentials (EPSPs) from Group la afferents reported in [80] and inhibitory postsynaptic potentials (IPSPs) from Group la inhibitory interneurons observed in [81].

The modelled SC did not include heteronymous excitatory projections for the sake of interpretability. Heteronymous projections in the human upper limb have been effectively reported in [82]. According to this study, inhibition occurs significantly more frequently than excitation between antagonist muscle pairs, with a higher frequency of inhibition received from antagonists muscles compared to muscles acting across a different joint. In [5, 83] it was noted that proximal to distal heteronymous Ia connections (from the arm to forearm) were absent, whilst wide connections from distal to proximal muscles were present, although weaker from proximal to shoulder muscles. The authors suggested that these stronger connections at wrist and elbow level may assist the hand muscles in grasping and lifting movements by providing stability to the corresponding joints. Consequently, heteronymous excitatory projections are reported to have less significance than heteronymous inhibitory projections in human upper limb movements. These findings are aligned with our SC model, in which reciprocal inhibition pathways allowed to capture this behaviour.

## 4 Methods

### 4.1 Ethics statement

Experiments were approved by the Commission cantonale d'éthique de la recherche sur l'être humain du canton de Vaud (CER-VD) under the license number 2017–02112 and performed in accordance with the Declaration of Helsinki. The two participants gave their written consent.

### 4.2 Spino-cerebellar control loop

The cerebellar and SC models operated in a closed loop with joint and muscle feedback (Fig 1A), in which cerebellar motor learning was assisted by fast reflex response and muscle activity regulation provided by the SC. The cerebellar model received sensory input signals describing the desired motor state (desired position, $Q_d$, and velocity, $\dot{Q}_d$, per joint) and the actual motor state of the upper limb musculoskeletal model (actual position, $Q_a$, and velocity, $\dot{Q}_a$, per joint). The comparison of the desired and actual motor states provided the instructive signal ($\epsilon$ per joint), also received by the cerebellum. The cerebellar output comprised a flexor-extensor (i.e., agonist-antagonist) pair of control signals ($M_f$ and $M_e$ per joint) that were sent to the SC model, which also received direct muscle feedback (length, $l_m$, and velocity, $\dot{l}_m$, per muscle). The SC generated the muscle excitation signals ($u_m$ per muscle) resulting in muscle activation which finally actuated the upper limb musculoskeletal model, thus closing the loop. To contextualise the spino-cerebellar integration, we also implemented the control loop lacking the SC circuits (Fig 1B). In this scenario, the cerebellar output signals were directly used as muscle excitation signals. The control loop included sensory and motor delays, mimicking the biological pathways. In the cerebellar sensorimotor pathway, there exists a delay ranging from about 100 to 150ms (with inter and intraindividual variations), accounting for the time spent from the generation of a motor command until sensing back its effect [84]. Regarding the SC to muscles transmission, a delay of about 30ms has been reported for the upper limb [85, 86]. Our spino-cerebellar model included a 50ms sensorial delay affecting the reception of sensory inputs in the cerebellum; a transmission delay of 30ms from the cerebellum to the SC, and

30ms from the SC to the muscles, total motor delay of 60ms. The asymmetry between sensory and motor delay stands for the higher latency found in neuromuscular junction, electromechanical and force generation delays (involved in the motor pathway), compared to the sensing, nerve conduction and synaptic delays (involved in the sensory pathway) [87].

The following subsections describe the different components of our spino-cerebellar control loop. The various building blocks were integrated using Robot Operating System (ROS), allowing a modular implementation.

## 4.3 Cerebellar model

We implemented a spiking neural network (SNN) replicating some cerebellar neural layers and equipped with spike-timing-dependent plasticity (STDP) to allow motor learning and adaptation. The cerebellar SNN model was adapted from previous models [18–21], which have already been used to study cerebellar motor learning, adaptation capabilities, and robustness to dynamic changes. Importantly, the cerebellar model presented here and the ones presented in the aforementioned previous works, all share the same neural populations and learning mechanism. Thus, building upon previous research, the current work focused on the influence of the SC on cerebellar motor learning.

The cerebellar SNN structure was divided in the following neural layers: i) mossy fibres (MFs), constituted the sensory input layer conveying the desired and actual motor state signals ($Q_d$, $\dot{Q}_d$, $Q_a$, $\dot{Q}_a$); ii) the spiking activity of MFs was transferred through excitatory afferents to the granule cell (GC) layer, where the sensory input information was univocally recoded; iii) the axons of the GCs, i.e., the parallel fibres (PFs), formed excitatory connections with the Purkinje cells (PCs); iv) PCs also received the excitatory action of the climbing fibres (CFs) conveying the instructive signal ($\epsilon$); v) the deep cerebellar nuclei (DCN) layer received the inhibitory action from PCs and excitatory connections from both MFs and CFs. The DCN spiking activity was translated into output motor commands (flexor-extensor motor control signals, $M_f$ and $M_e$) that constituted the cerebellar motor response to the sensory stimuli. Every neural layer was divided in two microcomplexes [72], being each microcomplex oriented to drive one of the two joints (shoulder or elbow). Each microcomplex at the PC-CF-DCN loop was partitioned into two regions: agonist and antagonist. The agonist region operated the joint flexor muscles, whereas the antagonist region operated the extensor muscles. This synergic agonist-antagonist (flexor-extensor) architecture allowed the cerebellar model to regulate the spatiotemporal muscle activity patterns [55], key for successful motor control [88]. See Fig 8 for a schematic representation of the cerebellar network, and Table 2 for network topology.

Consistently with the Marr-Albus-Ito theory on cerebellar motor adaptation [90–92], our cerebellar SNN model was equipped with synaptic plasticity at the GC-PC synapses. The synaptic weights were adjusted by means of an STDP mechanism that correlated the sensory information (univocally coded at GCs and transferred to PCs through PFs) and the instructive signal (conveyed to PCs by CFs). This STDP mechanism was a balanced process of long-term potentiation (LTP) and long-term depression (LTD). Each time a PC neuron received a GC spike through a PF, that synapse was potentiated (LTP) by a fixed amount as follows:

$$LTP\Delta W_{GC_i-PC_j}(t) = \alpha(\delta_{GCspike}(t) * dt) \tag{1}$$

where $\Delta W_{GC_i-PC_j}(t)$ stands for the synaptic weight change between GC $i$ and PC $j$; $\alpha = 0.006nS$ is the synaptic weight increment; and $\delta_{GCspike}(t)$ is the Dirac delta function of a GC spike, received at PCs through PFs.

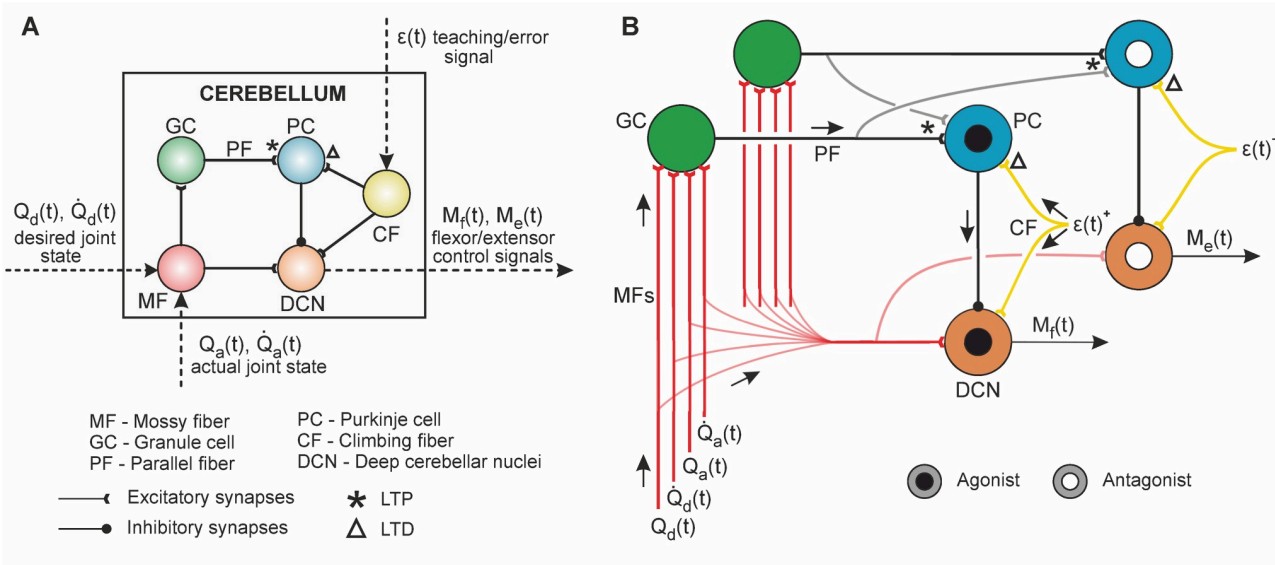

**Fig 8. Cerebellum model. A)** Neural layers, connections, input and output sensorimotor signals. The input signals are conveyed by the mossy fibres (MFs), which project excitatory synapses to the granule cells (GCs). These perform a recoding of the input signals, and project excitatory connections through the parallel fibres (PFs) reaching Purkinje cells (PCs). PF-PC connections are endowed with plasticity, balanced between the long-term potentiation (LTP) caused by the input PF spikes, and long-term depression derived from the climbing fibres (CFs) activity reaching PCs. CFs convey the instructive signal. Finally, PCs project inhibitory synapses towards the deep cerebellar nuclei (DCN), the output layer of the cerebellar model, which also receives a baseline excitatory action from MFs and CFs. **B)** Detailed schematic of the cerebellar connections. Each GC receives the input excitatory action from a unique combination of four MFs. Each input signal ($Q_d$, $\dot{Q}_d$, $Q_a$, $\dot{Q}_a$), is codified by ten MFs, being only one out of the ten MFs active at each time step. Hence, at each time step, four MFs will be active (one per input signal). That unique combination of four input MFs excites one single GC, allowing to perform a univocal representation of the sensory input at the granular layer. PCs then receive the excitatory action from all GCs in the cerebellar model and only one CF, allowing to relate the joint-specific instructive signal, to the global sensory state received from GCs. The PC-CF-DCN loop differentiates between agonist and antagonist regions, thus allowing simultaneous control of both flexor and extensor muscles.

**Table 2. Cerebellar neural topology.** Dashed entries stand for not applicable. Each GC-PC synapse was randomly initialised within the range [4.3, 5.2nS], except for one of the three repetitions of the 2000-trial motor adaptation process for each motor task. In that specific repetition, all GC-PC synapses were homogeneously initialised with 4.8nS[1].

| Neurons | | Synapses | | | |
|---|---|---|---|---|---|
| Pre-synaptic | Post-synaptic | Number | Type | Initial Weight (nS) | Weight range (nS) |
| 80 MFs | $20 \times 10^3$ GCs | $80 \times 10^3$ | AMPA | 0.18 | - |
| 80 MFs | 200 DCN | $16 \times 10^3$ | AMPA | 0.3 | - |
| $20 \times 10^3$ GCs | 200 PCs | $4000 \times 10^3$ | AMPA | rand [4.3, 5.2] | [0.0, 15.0] |
| 200 PCs | 200 DCN | 200 | GABA | 1.0 | - |
| 200 CFs | 200 PCs | 200 | AMPA | 0.0 | - |
| 200 CFs | 200 DCN | 200 | AMPA | 0.5 | - |
| 200 CFs | 200 DCN | 200 | NMDA | 0.25 | - |

[1] We performed an exhaustive search on the STDP parameters that govern the learning dynamics of both the spino-cerebellar and cerebellar models, as described in [16, 17, 89], since these parameters had the greatest impact on the models output commands. The selected parameters were chosen to cover the full working range of the different motor tasks. To ensure a fair comparison of the performance of the spino-cerebellar and cerebellar models, we used a common set of network parameters for both cases.

When the spiking activity of a CF conveyed an instructive signal to a PC neuron, the GC-PC connection that was involved in that error generation was depressed (LTD) as described by:

$$LTD\Delta W_{GC_i - PC_j}(t) = \beta * \int_{-\infty}^{t_{CFspike}} k(t - t_{CFspike}) * \delta_{GCspike}(t) * dt \qquad (2)$$

where $\beta = -0.003nS$ stands for the synaptic weight decrement; and $k(x)$ defines the integrative kernel with eligibility trace correlating past sensory inputs with the present instructive signal, i.e., the amount of LTD due to a CF spike depended on the previous GC activity received at PCs through PFs (see [20, 21] for a further description). A well-balanced LTP-LTD process changed the PF-PC synaptic weights, thus modifying the PCs output activity and the inhibitory action of PCs over DCN neurons, which ultimately varied the DCN output activity. Modulating the DCN activity allowed adaptation of the output motor response to the input stimuli. An iterative exposure to the sensory patterns defining the desired motor task, allowed adapting the motor response for error reduction.

We used leaky integrate and fire (LIF) neurons (see Supporting Information S1 Text) and EDLUT simulator [93] to build the cerebellar SNN model. Please see [20, 21] for a further review of the STDP mechanism and cerebellar layers.

## 4.4 Spinal cord model

Our SC model integrated the descending control signals from the cerebellum and the direct muscle feedback (Fig 9A). The SC model allowed fast reflex response and muscle activity regulation by means of monosynaptic Ia stretch reflex and disynaptic reciprocal inhibition pathways between antagonist muscles. The motoneuron (MN) of a given muscle received the following inputs: i) an excitatory connection conveying the cerebellar output signal ($M_f$ or $M_e$, for flexor or extensor muscle); ii) an excitatory connection from the Ia afferent fibre of the muscle (i.e., stretch reflex); iii) an inhibitory connection from the Ia interneuron (Ia IN) innervated by the Ia afferent of the antagonist muscle (i.e., reciprocal inhibition). The antagonist relation between the muscles of the upper limb model is detailed below. The neuron leaky integrate dynamics of the MN firing rate, $r$, were modelled as follows:

$$\tau \dot{r}(t) = -r(t) + \sigma(\sum_i w_i r_i(t - \tau_i)) \qquad (3)$$

where $\tau = 1ms$ stands for the spinal neuron activation time constant; $\sigma(x) = \frac{1}{1+exp(-D(x-0.5))}$ with $D = 8$, emulating the on-off behaviour of neurons; $i$ describes the MN input signals; $w_i$ is the synaptic weight of the input connection, being 1.0 for excitatory synapses and 0.5 for the inhibitory to reproduce physiological connectivity [80, 81] (see Discussion for further details); $r_i$ is the input activity; and $\tau_i = 30ms$ stands for the stretch reflex response delay. Depending on neuron size, $\tau$ can vary from 1 to 10ms [94], we only considered fast-response neurons as in [30]. For the upper limb, $\tau_i$ is about 30ms [85, 86]. The output rates of the MNs are finally provided as muscle excitation signals to the musculoskeletal model through a sigmoid ($u(t) = \sigma(r(t))$), thus inducing movement. The dynamics of Ia IN neurons followed the same description, with differing input activity including inhibitory connections between antagonist Ia IN (Fig 9B).

We used Prochazka's model for the Ia afferent feedback dynamics [95], with a mean firing rate of 10Hz [30, 96, 97]:

$$r_{Ia}(t) = sgn(\dot{l}_m(t)) * 4.3 |\dot{l}_m(t)|_+^{0.6} + 2(l_m(t) - l_{0,m}) + 10 \qquad (4)$$

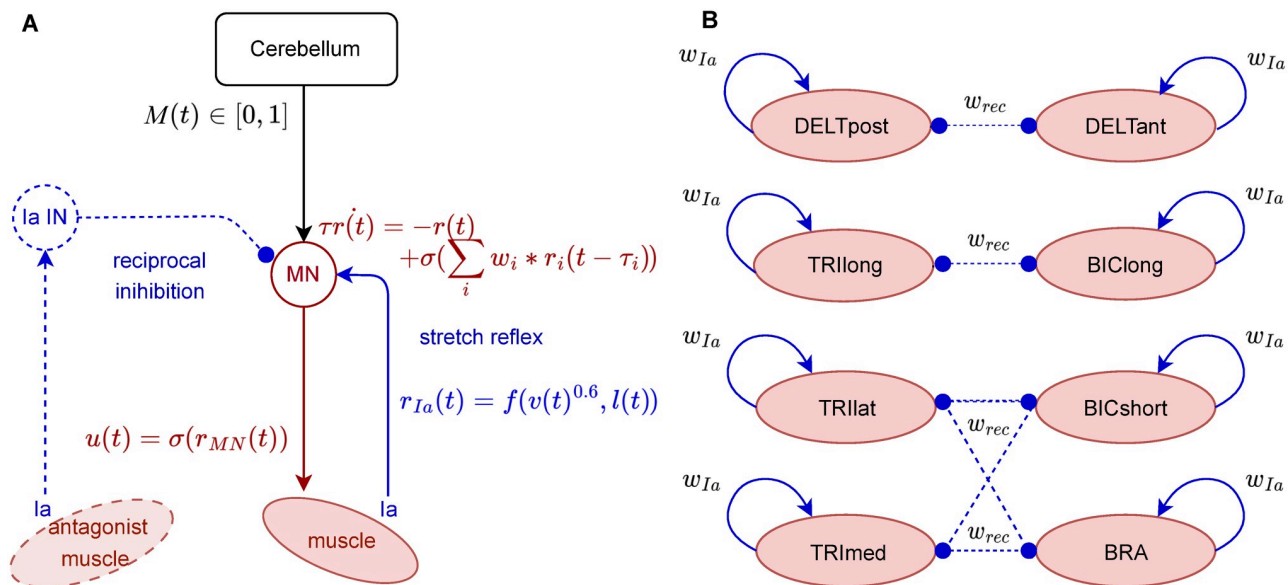

**Fig 9. Spinal cord model. A)** The spinal cord circuits were modelled as one motoneuron per muscle, receiving an excitatory input control signal (M) from the cerebellum, an excitatory connection from the Ia afferent fibre of the muscle (i.e., stretch reflex) and an inhibitory connection from the Ia interneuron (Ia IN) innervated by the Ia afferent of the antagonist muscle (i.e., reciprocal inhibition). We also included inhibitory connections between antagonist Ia interneurons. Each neuron is modelled with leaky integrate dynamics. **B)** Antagonist relation between the eight upper limb muscles: all the muscles shared the same synaptic weight for the stretch reflex and reciprocal inhibition pathways, i.e., 1.0 for excitatory synapses and 0.5 for the inhibitory.

where $l_m$ and $\dot{l}_m$ describes the muscle fibre length and velocity in mm and mm/s; and $|x|_+ = max(|x|, 0.01)$. The output rate, $r_{Ia}$, was scaled by its maximum $r_{Ia,max}$ to get a normalised value, i.e., $r_{Ia} \epsilon [0,1]$.

To model the SC we used FARMS Python library, developed at the BioRobotics laboratory [98].

## 4.5 Musculoskeletal upper limb model

We used a two DOF musculoskeletal upper limb model as the front-end body to be controlled. The model, adapted from [41], included two flexion-extension joints: shoulder and elbow. The model was actuated by eight Hill-based muscles [99], with the following joint distribution: i) for the shoulder, flexion was carried by the deltoid anterior (DELTant) and the biceps long (BIClong), and extension was conducted by the deltoid posterior (DELTpost) and the triceps long (TRIlong); ii) for the elbow, flexion was provided by the biceps long and short (BICshort) and the brachialis (BRA), whilst extension was allowed by the triceps long, lateral and medial (TRIlat, TRImed). Note that BIClong and TRIlong were biarticular muscles, as they actuated both joints. The antagonist relation between muscles is depicted in Fig 9B. The Hill-based muscle dynamics were the following:

$$\begin{cases} f_m = (a*f_{lv}(l_m, \dot{l}_m) + f_p(l_m))*cos\theta \\ \dfrac{da}{dt} = \dfrac{u - a}{\tau(u, A)} \end{cases} \quad (5)$$

with $f_m$ the muscle force, $f_{lv}$ a combination of the force-length and force-velocity curves, $f_p$ the

passive force-length curve, $\theta$ the pennation angle, $a$ the muscle activation (i.e., the concentration of calcium ions within the muscle), and $u$ the muscle excitation (i.e., the firing of the MN) [99]. We used OpenSim physics engine to simulate the muscle and skeleton dynamics [42]. To allow using kinematics and EMG from lab recordings, an OpenSim upper limb model was scaled to match the morphology of each lab participant. This scaling process was achieved using OpenPifPaf Human Pose Estimation algorithm [100] during the static period and OpenSim scaling tool.

## 4.6 Benchmarking with various motor tasks

We used a set of different motor tasks to be performed by the spino-cerebellar and cerebellar models, differentiating between two scenarios: lab recorded and lab designed motor tasks.

For the lab recorded scenario, we used kinematics and EMG recordings from healthy participants performing different arm movements. Experiments were approved by the Commission cantonale d'éthique de la recherche sur l'être humain du canton de Vaud (CER-VD) under the license number 2017–02112 and performed in accordance with the Declaration of Helsinki in NeuroRestore laboratory at Lausanne CHUV. After obtaining their written consent, two participants, P1 and P2, were asked to perform planar reaching movements (flexion-extension) and continuous circular movements, both movements performed in the vertical plane and at various speeds (self-selected speeds). For flexion-extension movements both shoulder and elbow moved in the same direction, whilst during the continuous circular movements the joints moved in opposite directions. Thus, our benchmark includes interaction torques both assisting and resisting the movement. The recorded kinematics (i.e., joint position and velocity) constituted the desired motor state ($Q_d$, $\dot{Q}_d$) used as the control loop sensory input, whilst the EMG recordings supported model validation in muscle space. For each recorded motor task we ran the experimental setup with both the spino-cerebellar and cerebellar models, using an OpenSim upper limb model scaled to match the participant's morphology. We then compared the models' experimental performance to the lab recordings in both joint and muscle spaces.

P1 and P2 movements were recorded using an RGB-D camera, and we used OpenPifPaf human pose estimation algorithm [100] to extract the 2D positions of the participant's anatomical joints at a frame rate of 25fps. Then 3D pose was deduced from the 2D pose, camera intrinsic, and depth information after accounting for distortion. The occlusions were removed using specially designed filters that ensure coherence in joint anatomy and time. We scaled an OpenSim upper limb musculoskeletal model to match the participant's morphology, and ran inverse kinematics (IK) over the body segment kinematics, thus allowing the extraction of joint position and velocity from the participant's motion. P1 generally performed fast movements, and the kinematics recordings of the fast circular movements were too noisy to extract joint position, thus we excluded this scenario from our analysis. For muscle activity, we recorded EMG using Delsys system and Trigno Avanti and Trigno Quattro sensors with an acquisition frequency of 1259.3Hz. We aligned the EMG with the kinematics signals thanks to a trigger inducing a pulse in an additional EMG channel and lightning a led in the camera range. We then computed the EMG envelopes to compare with our models muscle activation signals. For each recorded signal, we removed the mean and rectified the signal, which was then filtered using a low pass Butterworth filter with a 5Hz cutoff frequency. We applied the same processing steps to the maximal voluntary contraction (MVC) signal of each muscle (recorded at the beginning of the session), and used the maximal value of the processed MVC to normalise the corresponding muscle processed EMG signal.

For the lab designed scenario, we implemented a set of flexion-extension movements with different bell-shaped joint velocity profiles, characteristic of multi-joint arm reaching movements [44]. We then used the joint kinematics ($Q_d$, $\dot{Q}_d$) as the desired motor state to be performed by the spino-cerebellar and cerebellar models (please see Supporting Information for a depiction of the bell-shaped trajectories, S10–S12 Figs). We broadened the benchmark by adding a perturbation study using these bell-shaped trajectories. After cerebellar learning consolidation, we applied a set of motor perturbations whilst the trajectories were being performed: 50N for 30ms, applied to the hand in different directions and at different points along the flexion-extension movement. Each perturbation type was applied to 50 separate trials to get an average response, leaving 3 non-perturbed trials in between perturbed trials so that the model returned to its unperturbed state. Note that cerebellar learning was disabled during the perturbation study, to avoid cerebellar adaptation to the external forces and focus on SC response. To further extend the perturbation benchmark we tested two additional models in the aforementioned setup: i) SR-cerebellar model; i.e., spinal cord model equipped with just stretch reflex; ii) RI-cerebellar model; i.e., spinal cord model equipped with just reciprocal inhibition. This scenario allowed us to assess the influence of each implemented SC mechanism in handling perturbations.

Using this motor benchmark, and comparing the performance of the spino-cerebellar and cerebellar models, we could evaluate the cerebellum and spinal cord integration in terms of: muscle activity, motor adaptation and joint space performance, synaptic adaptation, and response to motor perturbations, for various trajectories with different initial and final positions and speeds. Please see Supporting Information for a representation of the motor tasks joint kinematics (S1–S12 Figs).

## 4.7 Cerebellar instructive signal

The cerebellar instructive signal $\epsilon(t)$ was obtained as the mismatch between the desired and actual joint state, combining in a single value per joint both position and velocity errors as follows:

$$\epsilon(t) = K_p[Q_d(t) - Q_a(t)] + K_v[\dot{Q}_d(t) - \dot{Q}_a(t)] \tag{6}$$

where $K_p$ = 3 and $K_v$ = 1 are the position and velocity error gain, respectively. The trajectory error signal in joint space can be derived from the proprioceptive and sensory information conveyed by the spino-cerebellar tract from the muscle spindles (muscle length) and Golgi tendon organs (muscle force) to the cerebellum [101].

Both the spino-cerebellar and cerebellar models were trained exclusively using kinematics to highlight the influence of the SC in modulating muscle activity. If EMG data had been incorporated in training the models, the cerebellar network would adapt and learn to replicate the recorded muscle patterns, resulting in similar muscle recruitment strategies for both the spino-cerebellar and cerebellar models. In other words, the influence of the SC in muscle activity modulation would be diminished.

## 4.8 Performance metrics

**4.8.1 Measuring kinematics performance.** To evaluate the kinematic performance of the spino-cerebellar and cerebellar models, we defined a set of metrics based on the mean absolute

error (MAE) between the desired $(Q_d, \dot{Q}_d)$ and actual $(Q_a, \dot{Q}_a)$ motor state of the arm:

$$
\begin{cases}
MAE_{pos}(t) = \dfrac{1}{N} \sum_{j=1}^{N} |Q_{d,j}(t) - Q_{a,j}(t)| \\[3mm]
MAE_{vel}(t) = \dfrac{1}{N} \sum_{j=1}^{N} |\dot{Q}_{d,j}(t) - \dot{Q}_{a,j}(t)|
\end{cases}
\tag{7}
$$

where $N$ stands for the number of joints (2), and $j$ for the joint index. We considered the position and velocity MAE of each motor task trial to assess the performance accuracy:

$$
\begin{cases}
MAE_{pos} = \dfrac{1}{T} \sum_{t=0}^{T} MAE_{pos}(t) \\[3mm]
MAE_{vel} = \dfrac{1}{T} \sum_{t=0}^{T} MAE_{vel}(t)
\end{cases}
\tag{8}
$$

where $T$ stands for the motor task period. We finally averaged these values over the last 200 trials of the motor adaptation process, and compared the final performance of the two models with the final mean $MAE_{pos}$ and $MAE_{vel}$ ($\bar{MAE}_{pos,f}$, $\bar{MAE}_{vel,f}$). We also computed the standard deviation ($std$) and the T-test p-value between the two models' performance with a T-test for the means of two independent samples of values [102] (computed using Python function scipy.stats.ttest_ind [103]).

**4.8.2 Measuring learning performance.** To measure the learning convergence (i.e., number of trials required to reach a stable trajectory tracking), we used control chart metrics [43]. Throughout the $MAE_{pos}$ and $MAE_{vel}$ curve of each motor task (all performed 3 times, each repetition consisting of 2000 learning trials) we computed the mean ($\mu$) and standard deviation ($\sigma$) using a sample size of 200 trials, which provided the following performance stability limits:

$$
\begin{cases}
L1 = \bar{MAE}_x \in [\mu - \sigma, \mu + \sigma] \\[2mm]
L2 = \bar{MAE}_x \in [\mu - 3\sigma, \mu - 2\sigma] \cup [\mu + 2\sigma, \mu + 3\sigma] \\[2mm]
L3 = \bar{MAE}_x \in ]-\infty, \mu - 3\sigma] \cup [\mu + 3\sigma, +\infty[
\end{cases}
\tag{9}
$$

We then checked the percentage of those 200 trials within each limit. As the limits were defined by the $std$, we also checked that the $std$ value was below 0.012rad for position and 0.055rad/s for velocity. Thus, at trial $x$, the behaviour was stable if the percentage of the 200 previous trials within each limit fulfilled the metrics defined in Table 3, and the $std$ was equal or below the aforementioned values. We then averaged each motor task metrics obtained from each of the three repetitions. By comparing the learning convergence of the spino-cerebellar and cerebellar models (i.e., number of trials required to reach a stable performance) we quantified the effect of the SC in the cerebellar motor adaptation process.

Additionally, we assessed the learning speed of the two models by considering the number of trials required to reach a target $MAE_{pos}$ of 0.1rad and a target $MAE_{vel}$ of 0.5rad/s. We defined

**Table 3.** *MAE convergence criteria from control chart.*

| Stability limit | $MAE_{pos}$ | $MAE_{vel}$ |
|---|---|---|
| $L1 = MAE_x \in [\mu - \sigma, \mu + \sigma]$ | $\geq 75\%$ | $\geq 73\%$ |
| $L2 = MAE_x \in [\mu - 3\sigma, \mu - 2\sigma] \cup [\mu + 2\sigma, \mu + 3\sigma]$ | $\leq 3\%$ | $\leq 3\%$ |
| $L3 = MAE_x \in ]-\infty, \mu - 3\sigma] \cup [\mu + 3\sigma, +\infty[$ | $\leq 2\%$ | $\leq 2\%$ |
| $\sigma$ | $\leq 0.012$ | $\leq 0.055$ |

the learning speed metric as 1 over this number of trials ($N_{trials}^{-1}$). The target values (0.1 rad and 0.5 rad/s) were such that they provided a common measure for all motor tasks whilst also taking into account the diversity of final MAE values amongst the different motor tasks. The final mean MAE of all motor tasks (including both spino-cerebellar and cerebellar models) was 0.03 rad for position, and 0.27 rad/s for velocity. To set the learning speed targets, we doubled these mean values and rounded them to the nearest tenth, resulting in 0.1 rad for position and 0.5 rad/s for velocity.

Thus, we evaluated how long it took for the performance to stabilise (learning convergence) and how fast the performance approached accurate tracking (learning velocity).

**4.8.3 Measuring cerebellar synaptic adaptation.**   To conduct a direct comparison between the synaptic adaptation of both the spino-cerebellar and cerebellar models, the GC-PC synaptic weights were homogeneously initialised in one of the three repetitions of the motor adaptation process for each motor task; thus providing a common synaptic starting point that allowed studying the differences at the synaptic level.

To study the effect of the SC in cerebellar synaptic adaptation we quantified the difference in the synaptic weight distribution at GC-PC connections between the spino-cerebellar and cerebellar models. Each PC was innervated by all GCs in the model; i.e., a GC formed an excitatory synapse with each PC (total number of GCs in the model $i$ = 20000; total number of PCs in the model $j$ = 200). We stored the synaptic weight of all GC-PC synapses in a matrix of size $i$x$j$:

$$W = \begin{bmatrix} w_{1,1} & w_{1,2} & \cdots & w_{1,j} \\ w_{2,1} & w_{2,2} & \cdots & w_{2,j} \\ \cdots & & & \\ w_{i,1} & w_{i,2} & \cdots & w_{i,j} \end{bmatrix} \qquad (10)$$

where $w_{x,y}$ is the synaptic weight of the synapse between GC $x$ and PC $y$.

We then represented the normalised weights stored in W, using $i$ as the y-axis and $j$ as the x-axis, providing a visual representation of the synaptic weight distribution (Fig 4A and 4B). To analyse the differences between the synaptic patterns that were formed in each model, we applied to the images Shannon's entropy from [104], thus providing a quantitative measure of the complexity of the synaptic distribution. The higher the entropy, the more heterogeneous the synaptic weights; i.e., more specialised GC-PC connections were formed.

To measure the number of GC neurons required by the spino-cerebellar and cerebellar models to successfully adapt to each motor task, we measured the percentage of GC-PC synapses that, by the end of the 2000 trials, had experienced a modification of their initial weight (set to 4.8nS, see Table 2).

**4.8.4 Measuring robustness against perturbations.**   To assess the robustness against perturbations, for each applied perturbation type we computed the mean MAE deviation from the no-perturbation scenario over the 50 perturbed trials as follow:

$$\begin{cases} \Delta\bar{MAE}_{pos} = \dfrac{1}{50} \sum_{i=1}^{50} |MAE_{pos,i} - \bar{MAE}_{pos,f}| \\ \\ \Delta\bar{MAE}_{vel} = \dfrac{1}{50} \sum_{i=1}^{50} |MAE_{vel,i} - \bar{MAE}_{vel,f}| \end{cases} \qquad (11)$$

where $MAE_{x,i}$ is the MAE resulting from the $i^{th}$ perturbed trial and $\bar{MAE}_{x,f}$ the final MAE for the corresponding no-perturbation scenario. We also computed the standard deviation. The

MAE deviation results of the four models (spino-cerebellar, cerebellar, SR-cerebellar, and RI-cerebellar) were compared using a Kruskal-Wallis H-test [105] (computed using Python function scipy.stats.kruskal [103]) to assess the overall difference amongst the four cases, followed by a Dunn test [106] (computed using Python function scikit-posthocs.posthoc_dunn [107]) to conduct pairwise tests. We also compared the mean joint CCI values of the four models during 50 trials for each of the three trajectories, performed without perturbations.

**4.8.5 Measuring muscle space performance.** We also evaluated performance in the muscle space using the lab recorded benchmark. Activation signals from models are commonly compared to EMG envelopes, but such comparisons are generally difficult to achieve due to scaling issues that hinder a direct analogy between the model and the real muscle dynamics; EMG signals are difficult to normalise and subject to measurement errors [108]. Besides, our musculoskeletal model is a simplification of the human upper limb, thus further hindering direct comparison of recorded EMG and the models muscle activation. To overcome this issue, we followed a more comprehensive approach by computing the correlation between activation signals and EMG envelopes; a commonly adopted solution to the aforementioned limitations [109–111]. We computed the EMG envelopes by rectifying and low pass filtering the signals using a 5th order Butterworth filter with a cut-off frequency of 5Hz. We also recorded the maximal velocity contraction (MVC) signals for each participant, we processed them the same way and finally normalised the EMG signals by the maximum of the muscle MVC signal. Then, for each movement type, we considered only the main activated muscles with clear activation patterns during the recordings, i.e., DELTant, BIClong, BICshort, TRIlat and BRA for P1 flexion-extension movements; DELTant, DELTpost, BIClong, TRIlat and BRA for P1 circular movements; DELTant, BIClong, TRIlong and TRIlat for P2 flexion-extension movements; and DELTant, DELTpost, and BRA for P2 circular movements. Thus, there is inter-participant variability in muscle patterns, as previously described for multi-joint movements in [112] (the participants' recorded EMG data and the corresponding main muscle patterns are displayed in Fig 5). Additionally, the differences in EMG strategies between participants were influenced by variations in movement kinematics, i.e., the participants did not perform the exact same movements (refer to Fig 2 and S1–S9 Figs for the joint kinematics of each P1 and P2 movement). It is worth noting that during flexion-extension movements, P2 exhibited smaller shoulder extension and larger elbow flexion compared to P1, resulting in greater activation of the BICshort and BRA muscles, and lesser activation of the DELTpost and TRIlong muscles. Similarly, during circular trajectories, P2 exhibited greater elbow flexion corresponding to larger activation of BICshort and BRA muscles. In our experimental setup, we computed the maximal correlation around lag 0 (on a window of one-fourth of the movement duration) for the 200 trials reaching the learning convergence metric and extracted the mean, standard deviation and T-test p-value between the spino-cerebellar and cerebellar model results. Regarding the lab recorded data, we did not consider those muscles that presented low and noisy EMG signals; however, those muscles were actually activated in our experimental simulations. Our musculoskeletal model indeed contained only eight muscles, so that such overactivation may reproduce other non-modelled muscle recruitment.

To study our cocontraction hypothesis, we computed and compared the cocontraction index (CCI) for each joint. From lab recordings or experimental simulations, we considered the average of EMG envelop or muscle activation signals, respectively, within each agonist and antagonist muscle group (i.e, DELTant and BIClong for shoulder flexor muscles; DELTpost and TRIlong for shoulder extensors; BIClong, BICshort and BRA for elbow flexors; TRIlong and TRIlat for elbow extensors). It is worth noting that biarticular muscles play specialised roles in energy-efficient transfer of momentum between joints [113], whilst also contributing to movement stabilisation through cocontraction [39]. Biarticular muscles were fully

accounted for when computing the CCI for both the shoulder and elbow. The joint CCI was computed using the method developed in [114], by which high CCI values correspond to high levels of activation of both agonist and antagonist muscle groups, and low CCI values indicate poor activation of both muscle groups, or high activation of one muscle group and low activation of the opposing group. Importantly, this method for extracting CCI was later evaluated in [115], demonstrating a strong correlation between CCI and joint stiffness. Joint CCI was given by:

$$CCI_j(t) = \frac{E\bar{M}G_{j,l}(t)}{E\bar{M}G_{j,h}(t)} \left( E\bar{M}G_{j,l}(t) + E\bar{M}G_{j,h}(t) \right) \tag{12}$$

where $E\bar{M}G_{j,l}$ is the level of activity in the less active muscle group and $E\bar{M}G_{j,h}$ the level of activity in the most active muscle group for each joint. As this index is also sensitive to scaling, we computed the maximal correlation around lag 0 (on a window of one-fourth of the movement duration) for the first 200 trials reaching our learning convergence metric (see Methods) and extracted the mean, standard deviation and T-test p-value between the spino-cerebellar and cerebellar model results. We also computed the mean joint CCI over each trajectory. A similar trend as that seen for the $MAE_{vel}$ was observed. We studied this potential relationship through a linear regression over all P1 and P2 trajectories.

## Supporting information

**S1 Fig. Spino-cerebellar and cerebellar models kinematics performance for the lab recorded scenario, participant 1 (P1) slow flexion-extension. A)** Position and velocity mean absolute error (MAE) over the 2000-trial motor adaptation process for both the spino-cerebellar and cerebellar models performing P1's slow flexion-extension (2.3 s). **B)** Joint kinematics of the first 200 trials (mean and standard deviation, std) for both models performing P1's slow flexion-extension (2.3 s). **C)** Joint kinematics of the last 200 trials (mean and std) for both models performing P1's slow flexion-extension (2.3 s).
(TIF)

**S2 Fig. Spino-cerebellar and cerebellar models kinematics performance for the lab recorded scenario, participant 1 (P1) moderate flexion-extension. A)** Position and velocity mean absolute error (MAE) over the 2000-trial motor adaptation process for both the spino-cerebellar and cerebellar models performing P1's moderate flexion-extension (1.7 s). **B)** Joint kinematics of the first 200 trials (mean and standard deviation, std) for both models performing P1's moderate flexion-extension (1.7 s). **C)** Joint kinematics of the last 200 trials (mean and std) for both models performing P1's moderate flexion-extension (1.7 s).
(TIF)

**S3 Fig. Spino-cerebellar and cerebellar models kinematics performance for the lab recorded scenario, participant 1 (P1) fast flexion-extension. A)** Position and velocity mean absolute error (MAE) over the 2000-trial motor adaptation process for both the spino-cerebellar and cerebellar models performing P1's fast flexion-extension (1.3 s). **B)** Joint kinematics of the first 200 trials (mean and standard deviation, std) for both models performing P1's fast flexion-extension (1.3 s). **C)** Joint kinematics of the last 200 trials (mean and std) for both models performing P1's fast flexion-extension (1.3 s).
(TIF)

**S4 Fig. Spino-cerebellar and cerebellar models kinematics performance for the lab recorded scenario, participant 1 (P1) moderate circle trajectory. A)** Position and velocity

mean absolute error (MAE) over the 2000-trial motor adaptation process for both the spino-cerebellar and cerebellar models performing P1's moderate circle trajectory (1.3 s). **B)** Joint kinematics of the first 200 trials (mean and standard deviation, std) for both models performing P1's moderate circle trajectory (1.3 s). **C)** Joint kinematics of the last 200 trials (mean and std) for both models performing P1's moderate circle trajectory (1.3 s).
(TIF)

**S5 Fig. Spino-cerebellar and cerebellar models kinematics performance for the lab recorded scenario, participant 2 (P2) slow flexion-extension. A)** Position and velocity mean absolute error (MAE) over the 2000-trial motor adaptation process for both the spino-cerebellar and cerebellar models performing P2's slow flexion-extension (3.5 s). **B)** Joint kinematics of the first 200 trials (mean and standard deviation, std) for both models performing P2's slow flexion-extension (3.5 s). **C)** Joint kinematics of the last 200 trials (mean and std) for both models performing P2's slow flexion-extension (3.5 s).
(TIF)

**S6 Fig. Spino-cerebellar and cerebellar models kinematics performance for the lab recorded scenario, participant 2 (P2) moderate flexion-extension. A)** Position and velocity mean absolute error (MAE) over the 2000-trial motor adaptation process for both the spino-cerebellar and cerebellar models performing P2's moderate flexion-extension (2.4 s). **B)** Joint kinematics of the first 200 trials (mean and standard deviation, std) for both models performing P2's moderate flexion-extension (2.4 s). **C)** Joint kinematics of the last 200 trials (mean and std) for both models performing P2's moderate flexion-extension (2.4 s).
(TIF)

**S7 Fig. Spino-cerebellar and cerebellar models kinematics performance for the lab recorded scenario, participant 2 (P2) slow circle trajectory. A)** Position and velocity mean absolute error (MAE) over the 2000-trial motor adaptation process for both the spino-cerebellar and cerebellar models performing P2's slow circle trajectory (2.7 s). **B)** Joint kinematics of the first 200 trials (mean and standard deviation, std) for both models performing P2's slow circle trajectory (2.7 s). **C)** Joint kinematics of the last 200 trials (mean and std) for both models performing P2's slow circle trajectory (2.7 s).
(TIF)

**S8 Fig. Spino-cerebellar and cerebellar models kinematics performance for the lab recorded scenario, participant 2 (P2) moderate circle trajectory. A)** Position and velocity mean absolute error (MAE) over the 2000-trial motor adaptation process for both the spino-cerebellar and cerebellar models performing P2's moderate circle trajectory (1.6 s). **B)** Joint kinematics of the first 200 trials (mean and standard deviation, std) for both models performing P2's moderate circle trajectory (1.6 s). **C)** Joint kinematics of the last 200 trials (mean and std) for both models performing P2's moderate circle trajectory (1.6 s).
(TIF)

**S9 Fig. Spino-cerebellar and cerebellar models kinematics performance for the lab recorded scenario, participant 2 (P2) fast circle trajectory. A)** Position and velocity mean absolute error (MAE) over the 2000-trial motor adaptation process for both the spino-cerebellar and cerebellar models performing P2's fast circle trajectory (1.2 s). **B)** Joint kinematics of the first 200 trials (mean and standard deviation, std) for both models performing P2's fast circle trajectory (1.2 s). **C)** Joint kinematics of the last 200 trials (mean and std) for both models performing P2's fast circle trajectory (1.2 s).
(TIF)

**S10 Fig. Spino-cerebellar and cerebellar models kinematics performance for the benchmark scenario, bell-shaped slow flexion-extension. A)** Position and velocity mean absolute error (MAE) over the 2000-trial motor adaptation process for both the spino-cerebellar and cerebellar models performing bell-shaped slow flexion-extension (3 s). **B)** Joint kinematics of the first 200 trials (mean and standard deviation, std) for both models performing bell-shaped slow flexion-extension (3 s). **C)** Joint kinematics of the last 200 trials (mean and std) for both models performing bell-shaped slow flexion-extension (3 s).
(TIF)

**S11 Fig. Spino-cerebellar and cerebellar models kinematics performance for the benchmark scenario, bell-shaped moderate flexion-extension. A)** Position and velocity mean absolute error (MAE) over the 2000-trial motor adaptation process for both the spino-cerebellar and cerebellar models performing bell-shaped moderate flexion-extension (2.3 s). **B)** Joint kinematics of the first 200 trials (mean and standard deviation, std) for both models performing bell-shaped moderate flexion-extension (2.3 s). **C)** Joint kinematics of the last 200 trials (mean and std) for both models performing bell-shaped moderate flexion-extension (2.3 s).
(TIF)

**S12 Fig. Spino-cerebellar and cerebellar models kinematics performance for the benchmark scenario, bell-shaped fast flexion-extension. A)** Position and velocity mean absolute error (MAE) over the 2000-trial motor adaptation process for both the spino-cerebellar and cerebellar models performing bell-shaped fast flexion-extension (1.5 s). **B)** Joint kinematics of the first 200 trials (mean and standard deviation, std) for both models performing bell-shaped fast flexion-extension (1.5 s). **C)** Joint kinematics of the last 200 trials (mean and std) for both models performing bell-shaped fast flexion-extension (1.5 s).
(TIF)

**S13 Fig. Spino-cerebellar and cerebellar models performance in muscle space for the recorded trajectories from P1.** The maximum correlation between activation signals and EMG around lag 0 are displayed for the main activated muscles during each movement type.
(TIF)

**S14 Fig. Spino-cerebellar and cerebellar models performance in muscle space for the recorded trajectories from P2.** The maximum correlation between activation signals and EMG around lag 0 are displayed for the main activated muscles during each movement type.
(TIF)

**S15 Fig. Evolution of the joint cocontraction index (CCI) over the motor adaptation process. A), B)** Joint CCI for both the spino-cerebellar and cerebellar models during the 2000-trial motor adaptation process for all P1 and P2 trajectories, respectively. Top row shows the shoulder CCI, bottom row displays the elbow CCI.
(TIF)

**S16 Fig. Spino-cerebellar and cerebellar models performance in kinematics space for bell-shaped trajectories.** The final performance (MAE), convergence time (from control chart), and learning speed (1 over the number of trials to reach a target MAE value) are compared.
(TIF)

**S17 Fig. Spino-cerebellar, SR-cerebellar, RI-cerebellar and cerebellar model response against external force perturbations during bell-shaped flexion-extension trajectories. A)** Position MAE deviation ($\Delta\bar{MAE}$) caused by all the perturbations applied during the 3s flexion-extension trajectory for the four models. Mean $\Delta\bar{MAE}$ and standard deviation (std) of 50

trials are displayed. **B)** Velocity MAE deviation ($\Delta \overline{MAE}$) caused by all the perturbations applied during the 3s flexion-extension trajectory for the four models. Mean $\Delta \overline{MAE}$ and std of 50 trials are displayed. **C)** Position MAE deviation ($\Delta \overline{MAE}$) caused by all the perturbations applied during the 1.5s flexion-extension trajectory for the four models. Mean $\Delta \overline{MAE}$ and std of 50 trials are displayed. **D)** Velocity MAE deviation ($\Delta \overline{MAE}$) caused by all the perturbations applied during the 1.5s flexion-extension trajectory for the four models. Mean $\Delta \overline{MAE}$ and std of 50 trials are displayed.
(TIF)

**S1 Text. Leaky integrate and fire neuron model dynamics.**
(DOCX)

**S1 Table. Neuron parameters.**
(XLSX)

## Acknowledgments

We gratefully thank the participants for their patience and willingness to collaborate in the recording sessions.

## Author Contributions

**Conceptualization:** Alice Bruel, Ignacio Abadía, Niceto R. Luque, Eduardo Ros, Auke Ijspeert.

**Data curation:** Alice Bruel, Ignacio Abadía.

**Formal analysis:** Alice Bruel, Ignacio Abadía.

**Funding acquisition:** Niceto R. Luque, Eduardo Ros, Auke Ijspeert.

**Investigation:** Alice Bruel, Ignacio Abadía, Thibault Collin, Icare Sakr.

**Methodology:** Alice Bruel, Ignacio Abadía.

**Software:** Alice Bruel, Ignacio Abadía.

**Supervision:** Henri Lorach, Niceto R. Luque, Eduardo Ros, Auke Ijspeert.

**Writing – original draft:** Alice Bruel, Ignacio Abadía.

**Writing – review & editing:** Alice Bruel, Ignacio Abadía, Thibault Collin, Icare Sakr, Henri Lorach, Niceto R. Luque, Eduardo Ros, Auke Ijspeert.

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
