## [Decision Letter · Decision Letter 0]

2 May 2023

Dear Mrs Bruel,

Thank you very much for submitting your manuscript "The spinal cord facilitates cerebellar upper limb motor learning and control; inputs from neuromusculoskeletal simulation" for consideration at PLOS Computational Biology.

As with all papers reviewed by the journal, your manuscript was reviewed by members of the editorial board and by several independent reviewers. In light of the reviews (below this email), we would like to invite the resubmission of a significantly-revised version that takes into account the reviewers' comments.

We cannot make any decision about publication until we have seen the revised manuscript and your response to the reviewers' comments. Your revised manuscript is also likely to be sent to reviewers for further evaluation.

Sincerely,

Aldo A Faisal

Academic Editor

PLOS Computational Biology

Daniele Marinazzo

Section Editor

PLOS Computational Biology

Reviewer's Responses to Questions

**Comments to the Authors:**

Reviewer #1: The authors employ a relatively simple model of spinal reflex pathways and a more elaborated model of cerebellar learning to test whether the spinal circuits enhance cerebellar learning of a range of limb movements compared to direct control of the muscles. All command signals to the spinal cord arise in the model of the cerebellum rather than in a model of motor cortex, which makes the system difficult to interpret in light of conventional distinctions between pyramidal and extrapyramidal control. This point needs better discussion, particularly in the context of the different phenotypes resulting from pathology in those subsystems.

The cerebellar model is organized by individual joints and the muscles that act upon them, rather than whole limb kinematics of the limb as proposed by Poppele and Bosco (2001; Physiological reviews, 2:539-568) from electrophysiological studies of the spinocerebellar tracts. The convergent inputs in the cerebellar parallel fibers and divergent outputs of the cerebellar deep nuclei seem incompatible with the joint-based architecture of this cerebellar model. Presumably this shows up in the clustering of synaptic weight distributions in Fig. 5A-B but this isn’t explained. Whether this unphysiological simplification affects the validity of the conclusions needs further discussion.

The modeled cerebellar output controls the motoneurons directly rather than modulating the gains of the homonymous and antagonist reflexes, which would be more consistent with the known connectivity and function, as now intimated in the Discussion. It is not clear how the arbitrarily chosen synaptic weights correspond to reflex gains. None of the well-known heteronymous excitatory stretch projections are modeled. Whether this affects the validity of the conclusions needs further discussion.

The authors discuss the importance of cocontraction but this is difficult to appreciate from the highly extracted cocontraction indices by joint, especially in view of the biarticular muscles that are known to have special functions related to energy-efficient transfer of momentum (van Ingen Schenau et al., 1994, Differential use and control of mono-and biarticular muscles. Human Movement Science, 13:495-517).

EMG data that are buried in the Supplemental Figures are confusing and concerning. For circular movements in S13B, subject P1 shows highly reciprocal EMG in anterior vs. posterior deltoid but very high cocontraction index for shoulder in D. For the same movements by P2 in Fig. S14, the EMGs show abnormally high cocontraction of anterior and posterior deltoids but lower shoulder CCI at the same point of 60-70% of movement cycle. The correlations between model muscle activations and recorded EMG from both P1 and P2 are modest for both models but said to all be statistically different between models. Given that the models were all trained on kinematics rather than EMG and the subjects have such different EMG strategies, that doesn’t seem possible.

The spinal circuitry model employed here is a tiny subset of the spinal circuitry that corresponds to aspects of servocontrol. Raphael et al. (2010; J. Neurosci. 30:9431-9444) showed that incorporation of even more of the known proprioceptive and fusimotor circuitry in a simpler model of a 2 DoF wrist improved the reinforcement learning of a range of tasks while reducing unphysiological cocontraction observed with servocontrol reflexes, the opposite of what was found here.

No attempt has been made to see how robust the learned cerebellar solutions are to small changes in the requested trajectory. Robust generalization is an important feature of biological learning that was demonstrated in Tsianos et al. (2014, cited).

It is difficult to assess the responses to perturbations (Figs. 7 & S18) in the absence of data about how human subjects respond to similar perturbations. Given the absence of any stretch reflex circuitry in the cerebellar model, it is not surprising that its kinematic responses to perturbations are larger. It isn’t clear, however, whether these responses are dominated by such delayed reflex responses or by cocontraction and intrinsic muscle properties that result in corrective forces with zero delay. Not allowing plasticity during any applied perturbations means that we have no data about how the models adapt to changing task conditions, which is likely to be an important role for the cerebellum.

Fig. 1: The anatomical drawings are too small to see the actions of the modeled muscles and their actions at the shoulder are described incorrectly in the caption – forward motion of the upper arm should be defined as flexion (e.g. anterior deltoid) and backward is extension (posterior deltoid). Their actions are described differently in Methods 4.3 but also incorrectly.

Fig. 2 legend claims all comparisons in C have p-value <0.001 but text says fast circle comparison is insignificant, as it clearly appears in the figure (which is labeled 1.3s but text says 1.2s). Slow circle (1.8s) doesn’t look significant, either. The fluctuating levels of velocity error late in the spino-cerebellar model look like over-training effects. Fig. 3 legend has similar inconsistencies. The authors would do well to use the convention of brackets and astirisks to indicate significance of each comparison rather than summarizing these in the figure legends and text.

Reviewer #2: Here the authors present a spino-cerebellar model of motor learning with the aim of understanding how the organising principles of the spinal cord can facilitate motor learning.

This model assumes that the Cerebellum is implementing an inverse model and uses this to generate a desired trajectory, while using the mismatch between the generated trajectory and the desired trajectory as a learning signal. However, Spinocerebellum is generally considered to modulate, not generate, descending motor commands to the spinal command (c.f. Krakauer, John W., et al. "Motor learning." Compr Physiol 9.2 (2019): 613-663., and Cerebellum chapter of Kandel, Eric R., et al., eds. Principles of neural science. Vol. 4. New York: McGraw-hill, 2000.). Furthermore, while there is evidence for inverse models in the Cerebellum, there is arguably more extensive evidence for a forward model (Izawa, Jun, Sarah E. Criscimagna-Hemminger, and Reza Shadmehr. "Cerebellar contributions to reach adaptation and learning sensory consequences of action." Journal of Neuroscience 32.12 (2012): 4230-4239. & Krakauer, John W., et al. "Motor learning." Compr Physiol 9.2 (2019): 613-663.), whereby the Cerebellum attempts to predict sensory consequences of descending motor commands, which in turn provides sensory prediction errors that drives learning (Tseng, Ya-weng, et al. "Sensory prediction errors drive cerebellum-dependent adaptation of reaching." Journal of neurophysiology 98.1 (2007): 54-62.). Given this, it is debatable how well this model reflects what we currently know about Cerebellar computation, and it would be useful if the authors could provide further justifications for the assumptions made in their model.

It could also be argued that certain findings described here are unsurprising: it seems self-evident that EMG/CCI data from human subjects is better matched by a model with reflex loops that are known to heavily shape coordinated muscle activity in natural movement, and we would fully expect greater robustness to perturbations when co-contraction mechanisms are hard wired by reciprocal inhibition and reflex loops rather than allowing the cerebellar model to try and learn them. Nonetheless these observations do provide useful emphasis on the importance of including such mechanisms when building models of sensorimotor loop. The significance of the paper would be improved if a deeper analysis was provided for the more notable/novel findings regarding improved motor learning rates and simpler solutions in the synaptic weight space when a spinal cord interface is included.

Additional comments:

i) Was any optimisation of network parameters required? If so, how was this performed?

ii) Having 2 separate figures (fig 2 and 3) for each subject seems excessive. Can you combine these two into a single figure?

iii) Why 0.1 rad and 0.5 rad/s for convergence targets?

iv) Panel A in all of figures 2-4 should ideally have a shaded region around the mean line to show standard deviation of the learning curves.

v) “To study learning convergence we applied control charts on the MAE data to determine the number of trials required to achieve a stable performance [39].” The definition of learning convergence could do with a more thorough description here (just one extra sentence better describing how to is calculated).

vi) Figure 4B-C should have error bars or std dev bars and statistical tests should be performed to confirm learning convergence and learning rates are significantly different for Cb-SC vs Cb-only models.

vii) Figure 5C,D could be more effectively visualised as % change in synaptic entropy relative to entropy at trial 0, and accompanied with a second plot with trials on the x axis, and difference between synaptic entropy in the Cb-SC vs Cb-only models on the y-axis. Then you can bin the Flex-ext and Circle tasks together into a one % change plot and one entropy difference plot for each task.

viii) Figure 5A,B could be used for a more interesting analyses, as it is not particularly surprising that the synaptic weights change over trials. What about checking if Cb-SC and Cb-only models converge to the same synaptic solutions every time, and if they converge to the same solutions as each other?

ix) Figures should more clearly label which plots correspond to which subjects.

x) Figure 6A/B could be more informative if the y-axis is the absolute difference between the muscle activation signal and the EMG data.

xi) How dependent is perturbation stability on the reflex vs reciprocal inhibition parts of the SC module?

**Have the authors made all data and (if applicable) computational code underlying the findings in their manuscript fully available?**

Reviewer #1: Yes

Reviewer #2: Yes

PLOS authors have the option to publish the peer review history of their article (what does this mean?). If published, this will include your full peer review and any attached files.

Reviewer #1: No

Reviewer #2: No
---

## [Decision Letter · Decision Letter 1]

13 Nov 2023

Dear Mrs Bruel,

Thanks for your patience and sorry for the delay. We are likely to accept this manuscript for publication, after giving you the chance to consider the remaining recommendation from Dr. Loeb concerning some more details and specifications in the discussion. I can personally guarantee that the turnover for this additional round will be very short, once the paper gets back to me.

Sincerely,

Daniele Marinazzo

Section Editor

PLOS Computational Biology

Reviewer's Responses to Questions

**Comments to the Authors:**

Reviewer #1: The authors have substantially revised the Ms and generated detailed responses to the reviewers.

This model is based on Ito’s 1997 presumption that an internal inverse model of the musculoskeletal plant can compute the motor commands required to achieve a desired behavioral goal. As recently pointed out, the oculomotor system for which this concept was derived is fundamentally different from most of the rest of the musculoskeletal system (Loeb, G. E., 2021, Learning to use Muscles, Journal of Human Kinetics, 76, 9). To their credit, the authors have included a small subset of the multiarticular muscles and dynamics, proprioceptive afferents and spinal circuitry that constitute this difference, but it remains unclear if the simplistic concept of joint-based microcomplexes will survive the larger reality. That reality is substantially underestimated by claiming that the spinal cord represents a “modification of the arm plant dynamics” (lines 499-500), particularly when considering that most of the cerebellar and cortical output is mixed with segmental sensory feedback in spinal interneurons before arriving at motoneurons. The authors’ cerebellocentric view of sensorimotor control begs the question of where it interacts with the inevitable (but still unclear) contributions of the pyramidal cortical system. In addition to SC, a large part of that interaction is probably also in the reticular formation and deep cerebellar nuclei to which both project, a complication mostly ignored also by those who study and model cerebral cortex. A bit more discussion of these complexities in this already rather long paper might help cerebral and cerebellar researchers to stop talking past each other.

**Have the authors made all data and (if applicable) computational code underlying the findings in their manuscript fully available?**

Reviewer #1: Yes

PLOS authors have the option to publish the peer review history of their article (what does this mean?). If published, this will include your full peer review and any attached files.

Reviewer #1: **Yes: **Gerald E. Loeb

Figure Files:

Data Requirements:

Reproducibility:

References:

---

## [Decision Letter · Decision Letter 2]

12 Dec 2023

Dear Mrs Bruel,

We are pleased to inform you that your manuscript 'The spinal cord facilitates cerebellar upper limb motor learning and control; inputs from neuromusculoskeletal simulation' has been provisionally accepted for publication in PLOS Computational Biology.

Best regards,

Aldo A Faisal

Academic Editor

PLOS Computational Biology

Daniele Marinazzo

Section Editor

PLOS Computational Biology

Reviewer's Responses to Questions

**Comments to the Authors:**

Reviewer #1: The additions provide better context for the reader.

**Have the authors made all data and (if applicable) computational code underlying the findings in their manuscript fully available?**

Reviewer #1: Yes

PLOS authors have the option to publish the peer review history of their article (what does this mean?). If published, this will include your full peer review and any attached files.

Reviewer #1: **Yes: **Gerald E Loeb

---

## [Editor Report · Acceptance letter]

22 Dec 2023

PCOMPBIOL-D-23-00370R2 

The spinal cord facilitates cerebellar upper limb motor learning and control; inputs from neuromusculoskeletal simulation

Dear Dr Bruel,

I am pleased to inform you that your manuscript has been formally accepted for publication in PLOS Computational Biology. Your manuscript is now with our production department and you will be notified of the publication date in due course.

With kind regards,

Zsofia Freund
